# PDETime: Rethinking Long-term Multivariate Time Series Forecasting from the Perspective of Partial Differential Equations

## Abstract

Recent advancements in deep learning have led to the development of various approaches for long-term multivariate time-series forecasting (LMTF). Most of these approaches can be categorized as either historical-value-based methods, which rely on discretely sampled past observations, or time-index-based methods that model time indices directly as input variables. However, real-world dynamical systems often exhibit nonstationarity and suffer from insufficient sampling frequency, posing challenges such as spurious correlations between time steps and difficulties in modeling complex temporal dependencies. In this paper, we treat multivariate time series as data sampled from a continuous dynamical system governed by partial differential equations (PDEs) and propose a new model called PDETime. Instead of predicting future values directly, PDETime employs an encoding-integration-decoding architecture: it predicts the partial derivative of the system with respect to time (i.e., the first-order difference) in the latent space and then integrates this information to forecast future series. This approach enhances both performance and stability, especially in scenarios with extremely long forecasting windows. Extensive experiments on seven diverse real-world LMTF datasets demonstrate that PDETime not only adapts effectively to the intrinsic spatiotemporal nature of the data but also sets new benchmarks by achieving state-of-the-art results.

## 1 Introduction

Multivariate time series forecasting plays a pivotal role in diverse applications, such as weather prediction (Angryk et al., 2020), energy consumption (Demirel et al., 2012), healthcare (Matsubara et al., 2014), and traffic flow estimation (Li et al., 2017). Generally, time series forecasting models can be roughly classified into two categories: historical-value-based models (Zhou et al., 2021; Wu et al., 2021; Zeng et al., 2023; Nie et al., 2023), and time-index-based models (Woo et al., 2023; Naour et al., 2023). The former predicts future time steps by leveraging historical observations, characterized by $\hat{\mathbf{x}}_{t+1} = \mathbf{F}_\theta(\mathbf{x}_t, \mathbf{x}_{t-1}, ...)$, while the latter solely utilizes the corresponding time-index features, denoted as $\hat{\mathbf{x}}_{t+1} = \mathbf{F}_\theta(t + 1)$. Historical-value-based models have gained popularity due to their simplicity and effectiveness, positioned as state-of-the-art in multivariate time series forecasting. However, it is crucial to acknowledge that multivariate time series data are often discretely sampled from continuous dynamical systems. This characteristic poses a challenge for historical-value-based models in LMTF, as they tend to capture spurious correlations limited to the insufficient sampling frequency (Gong et al., 2017; Woo et al., 2023).

Alternatively, deep time-index-based methods have garnered a significant amount of attention (Woo et al., 2023; Naour et al., 2023). These methods inherently address the limitations of historical-value-based methods by mapping the time-index features to target predictions in the continuous space through implicit neural representations (INRs) (Tancik et al., 2020; Sitzmann et al., 2020). While time-index-based models implicitly leverage historical observations to enhance their exploratory capabilities, they are primarily characterized by time-index coordinates. This limitation hiders their effectiveness in capturing complex temporal dependencies, resulting in performance that falls slightly behind that of historical-value-based models.

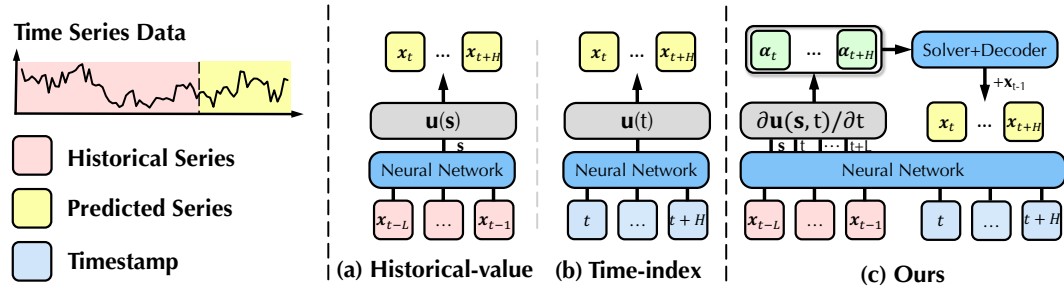

Figure 1: Comparison between historical-value-based models, time-index-based models and ours.

In this work, we introduce a novel perspective by framing multivariate time series as temporal data discretely sampled from a continuous dynamical system which is governed by partial differential equations (PDEs) as defined in Eq 1 (see Sec 3.1). From the PDEs perspective, illustrated in Figure 1, existing historical-value-based methods typically extract the underlying latent variables (denoted by $\mathbf{s}$), such as the position and physical properties of sensors which cannot be observed directly (which is also referred to as spatial information for the convenience of presentation). These models then predict future series with another network, formulated as $[\mathbf{x}_t, ..., \mathbf{x}_{t+L}] = \mathbf{u}_\theta(\mathbf{s})$, which neglects the temporal information. Conversely, time-index-based models focus solely on the time-index coordinates without explicitly incorporating spatial information, expressed as $\mathbf{x}_t = \mathbf{u}_\theta(t)$. It is evident that both the above models overlook either temporal or spatial information, making them incapable of modeling $\mathbf{u}(\mathbf{s}, t)$ as required by Eq 1, ultimately limiting their performance. Furthermore, as shown in Figure 1(c), instead of treating LMTF as easily input-output mapping learning by neural networks, which ignores the dependencies across predicted time steps. We propose to predict $\frac{\partial \mathbf{u}(\mathbf{s}, t)}{\partial t}$ instead of $\mathbf{u}(\mathbf{s}, t)$, and then generate $\mathbf{x}_t$ via the integral $\mathbf{x}_{t_0} + \int_{t_0}^{t} \frac{\partial \mathbf{u}(\mathbf{s}, \mu)}{\partial \mu} d\mu$, which implicitly capture temporal dependencies.

Motivated by the limitations of existing approaches and inspired by neural Solvers, we propose PDETime, a PDE-based model for long-term multivariate time-series forecasting (LMTF). PDETime employs an encoding-integration-decoding architecture and frames LMTF as an Initial Value Problem, explicitly incorporating both spatial and temporal information and leveraging numerical solvers. Specifically, PDETime initiates its process with a single initial condition, denoted as $\mathbf{x}_{t_0}$, and leverage neural networks to project the system's dynamics forward in time with three distinct steps. Firstly, PDETime generates the partial derivative term $E_\theta(\mathbf{X}_{his}, \mathbf{c}_t, \tau_t) = \boldsymbol{\alpha}_t \approx \frac{\partial \mathbf{u}(\mathbf{s}, t)}{\partial t}$ utilizing an encoder in latent space. Unlike traditional PDE problems, the spatial information $\mathbf{s}$ (latent variable) of LMTF is unknown. Therefore, the encoder estimates $\mathbf{s}$ based on historical observations. Subsequently, a numerical solver is employed to compute the integral term $\mathbf{z}_t = \int_{t_0}^{t} \boldsymbol{\alpha}_\mu d\mu$. The proposed solver effectively mitigates the accumulation error issue and enhances the stability of the prediction results compared to traditional Neural ODE solvers (Chen et al., 2018). In the final step, PDETime employs a decoder to translate the integral term from the latent space back to the value space, predicting the results as $\hat{\mathbf{x}}_t = \mathbf{x}_{t_0} + D_\phi(\mathbf{z}_t)$. Similar to time-index-based models, PDETime utilizes meta-optimization to enhance its ability to extrapolate across the forecast horizon. Additionally, PDETime can be simplified into either a historical-value-based or time-index-based model by omitting either the temporal or spatial domains, respectively.

In summary, the key contributions of this work are as follows:

- We present a novel perspective for LMTF by considering time series as data regularly sampled from a dynamical system governed by PDEs along the temporal domains.

- We propose PDETime, a PDE-based model inspired by neural Solvers, which tackles LMTF as an Initial Value Problem of PDEs. PDETime incorporates encoding-integration-decoding operations and leverages meta-optimization to extrapolate future series.

- We extensively evaluate the proposed model on seven real-world benchmarks across multiple domains under the long-term setting. Our empirical studies demonstrate that PDETime consistently achieves state-of-the-art performance. Moreover, PDETime has better performance

and stability, particularly in scenarios with extremely long forecasting windows, thanks to its encoding-integration-decoding architecture.

## 2 RELATED WORK

**Multivariate Time Series Forecasting.** With the progressive breakthrough made in deep learning, deep models have been proposed to tackle various time series forecasting applications. Depending on whether temporal or spatial is utilized, these models are classified into historical-value-based (Zhou et al., 2021; 2022; Zeng et al., 2023; Nie et al., 2023; Zhang & Yan, 2023; Liu et al., 2024; 2022b;a; Wu et al., 2023), and time-index-based models (Woo et al., 2023). Historical-value-based models, predicting target time steps utilizing historical observations, have been extensively developed and made significant progress in which a large body of work that tries to apply Transformer to forecast long-term series in recent years (Wen et al., 2023). Early works like Informer (Zhou et al., 2021) and LongTrans (Li et al., 2019) were focused on designing novel mechanism to reduce the complexity of the original attention mechanism, thus capturing long-term dependency to achieve better performance. Afterwards, efforts were made to extract better temporal features to enhance the performance of the model (Wu et al., 2021; Zhou et al., 2022). Recent work (Zeng et al., 2023) has found that a single linear channel-independent model can outperform complex transformer-based models. Therefore, the very recent channel-independent models like PatchTST (Nie et al., 2023) and DLinear (Zeng et al., 2023) have become state-of-the-art. In contrast, time-index-based models (Woo et al., 2023; Fons et al., 2022; Jiang et al., 2023; Naour et al., 2023) are a kind of coordinated-based models, mapping coordinates to values, which was represented by INRs. These models have received less attention and their performance still lags behind historical-value-based models. PDETime, unlike previous works, considers multivariate time series as spatiotemporal data and approaches the prediction target sequences from the perspective of partial differential equations.

**Implicit Neural Representations.** Implicit Neural Representations are the class of works representing signals as a continuous function parameterized by multi-layer perceptions (MLPs) (Tancik et al., 2020; Sitzmann et al., 2020) (instead of using the traditional discrete representation). These neural networks have been used to learn differentiable representations of various objects such as images (Henzler et al., 2020), shapes (Liu et al., 2020; 2019), and textures (Oechsle et al., 2019). However, there is limited research on INRs for times series (Fons et al., 2022; Jiang et al., 2023; Woo et al., 2023; Naour et al., 2023; Jeong & Shin, 2022). And previous works mainly focused on time series generation and anomaly detection (Fons et al., 2022; Jeong & Shin, 2022). DeepTime (Woo et al., 2023) is the work designed to learn a set of basis INR functions for forecasting, however, its performance is worse than historical-value-based models. In this work, we use INRs to represent spatial domains and temporal domains.

**Neural PDE Solvers.** Neural PDE solvers which are used for temporal PDEs, are laying the foundations of what is becoming both a rapidly growing and significant area of research. These neural PDE solvers fall into two broad categories, *neural operator methods* and *autoregressive methods*. The neural operator methods (Kovachki et al., 2021; Li et al., 2020; Lu et al., 2021) treat the mapping from initial conditions to solutions as time $t$ as an input-output mapping learnable via supervised learning. For a given PDE and given initial conditions $u_0$, the neural operator $\mathcal{M}$ is trained to satisfy $\mathcal{M}(t, \mathbf{u}_0) = \mathbf{u}(t)$ (historical-value-based and time-index-based models both can be seen as neural operator methods). However, these methods are not designed to generalize to dynamics for out-of-distribution $t$. In contrast, the autoregressive methods (Bar-Sinai et al., 2019; Greenfeld et al., 2019; Hsieh et al., 2019; Yin et al., 2022; Brandstetter et al., 2021; Lippe et al., 2024) solve the PDEs iteratively. The solution of autoregressive methods at time $t + \Delta t$ as $\mathbf{u}(t + \Delta) = \mathcal{A}(\mathbf{u}(t), \Delta t)$. In this work, We consider multivariate time series as data sampled from a continuous dynamical system according to a regular time discretization, which can be described by partial differential equations. For the given initial condition $\mathbf{x}_{t_0}$, PDETime use the numerous solvers (e.g., the Euler solver) to simulate target time step $\mathbf{x}_t$ which is more like autoregressive methods.

## 3 METHOD

### 3.1 PROBLEM FORMULATION

In contrast to previous works (Zeng et al., 2023; Woo et al., 2023), we regard multivariate time series as the spatio-temporal data regularly sampled from partial differential equations along the temporal

domain, denoted as $\mathbf{u}(\mathbf{s}, t)$, which satisfies the PDE equation:

$$\mathcal{F}(\mathbf{u}, \frac{\partial \mathbf{u}}{\partial t}, \frac{\partial \mathbf{u}}{\partial \mathbf{s}^1}, ..., \frac{\partial^2 \mathbf{u}}{\partial t^2}, \frac{\partial^2 \mathbf{u}}{\partial \mathbf{s}^2}, ...) = 0, \ \mathbf{u}(\mathbf{s}, t) : \Omega \times \mathcal{T} \to \mathcal{V}, \tag{1}$$

subject to initial and boundary conditions. Here $\mathbf{u}(\mathbf{s}, t)$ represents the spatio-temporal dependent and multi-dimensional continuous vector field, where $\Omega \in \mathbb{R}^C$ and $\mathcal{T} \in \mathbb{R}$ denote the spatial and temporal domains, respectively. For multivariate time series data, we regard attributes of sensors and external factors as spatial information (e.g., the position and physical properties of sensors) $\mathbf{s}$, which cannot be directly observed and can only be inferred from historical observations. On the other hand, the value of the temporal domains, $t$, is known and can include calendar information $\mathbf{c}$ associated with the time series data. LMTF is treated as an initial value problem in PDETime, where the objective is to infer $\mathbf{u}(\mathbf{s}, t) \in \mathbb{R}^C$ at a future time $t$ based on the known values $\mathbf{u}(\mathbf{s}, t_0)$. Consequently, this is achieved by utilizing the following formula:

$$\mathbf{u}(\mathbf{s}, t) = \mathbf{u}(\mathbf{s}, t_0) + \int_{t_0}^{t} \frac{\partial \mathbf{u}(\mathbf{s}, \mu)}{\partial \mu} d\mu. \tag{2}$$

PDETime initiates its process with a single initial condition, denoted as $\mathbf{u}(\mathbf{s}, t_0)$, and leverages neural networks to project the system's dynamics forward in time. The procedure unfolds in three distinct steps. Firstly, PDETime generates a latent vector, $\alpha_t$ of a predefined dimension $d$, utilizing an encoder function, $E_\theta : \Omega \times \mathcal{T} \to \mathbb{R}^d$ (denoted as the ENC step). Subsequently, it employs an Euler solver, a numerical method, to approximate the integral term, $\mathbf{z}_t = \int_{t_0}^{t} \alpha_\mu d\mu$, effectively capturing the system's evolution over time (denoted as the SOL step). In the final step, PDETime translates the latent vectors, $\mathbf{z}_t$, back into the spatial domain using a decoder, $D_\phi : \mathbb{R}^d \to \mathcal{V}$ to reconstruct the value space (denoted as the DEC step). This results in the following model, are illustrated in Figure 2,

$$(ENC) \ \alpha_t = E_\theta(\mathbf{X}_{his}, \mathbf{c}_t, \tau_t), \tag{3}$$

$$(SOL) \ \mathbf{z}_t = \int_{t_0}^{t} \alpha_\tau d\tau, \tag{4}$$

$$(DEC) \ \hat{\mathbf{x}}_t = D_\phi(\mathbf{z}_t) + \mathbf{x}_{t_0}. \tag{5}$$

We describe the details of the components in Section 3.2 and see Algorithm 3 for the training procedure of PDETime.

## 3.2 COMPONENTS OF PDETIME

### 3.2.1 ENCODER: $\alpha_t = E_\theta(\mathbf{X}_{his}, \mathbf{c}_t, \tau_t)$

The Encoder component computes the latent vector $\alpha_t$ representing the temporal derivative $\frac{\partial \mathbf{u}(\mathbf{s}, t)}{\partial t}$ of unknown field $\mathbf{u}(\mathbf{s}, t)$. Due to the unavailability of $\mathbf{u}(\mathbf{s}, t)$, it is not possible to directly ensure $\alpha_t = \frac{\partial \mathbf{u}(\mathbf{s}, t)}{\partial t}$. However, through Eq 13, it is observed that $\alpha_t$ is proportional to $\frac{\partial \mathbf{u}(\mathbf{s}, t)}{\partial t}$ when $\mathcal{L}_f \to 0$ and $\Delta t \to 0$ (See sec A.2 for more details). The Encoder leverages this observation to estimate temporal derivative effectively. In addition, the encoder utilizes historical observations $\mathbf{X}_{his}$ to extract the latent variable as the spatial information $\mathbf{s}$. Next, we briefly introduce the structure of the Encoder. In our Encoder, we employ Concatenated Fourier Features (CFF) (Woo et al., 2023; Tancik et al., 2020) and SIREN (Sitzmann et al., 2020) with $k$ layers to represent the high-frequency components of $\tau_t$, $\mathbf{X}_{his}$, and $\mathbf{c}_t$.

$$\tau_t^{(i)} = \text{GELU}(\mathbf{W}_\tau^{(i-1)} \tau^{(i-1)} + \mathbf{b}_\tau^{(i-1)}),$$
$$\mathbf{c}_t^{(i)} = \sin(\mathbf{W}_c^{(i-1)} \mathbf{c}^{(i-1)} + \mathbf{b}_c^{(i-1)}),$$
$$\mathbf{X}^{(i)} = \sin(\mathbf{W}_x^{(i-1)} \mathbf{X}^{(i-1)} + \mathbf{b}_x^{(i-1)}), \ i = 1, ..., k \tag{6}$$

where $\mathbf{X}^{(0)} \in \mathbb{R}^{L \times C} = \mathbf{X}_{his} = [\mathbf{x}_{t_0-L+1}, ..., \mathbf{x}_{t_0}]$, $\mathbf{c}_t^{(0)} \in \mathbb{R}^m$ is the temporal feature, and $\tau_t^{(0)} \in \mathbb{R}$ is the time-index feature where $\tau_t = \frac{t}{H+L}$ for $t = 0, 1, ..., H + L$, $L$ and $H$ are the look-back and horizon length, respectively. CFF is used to represent $\tau_t^{(0)}$, i.e. $\tau_t^{(0)} =$

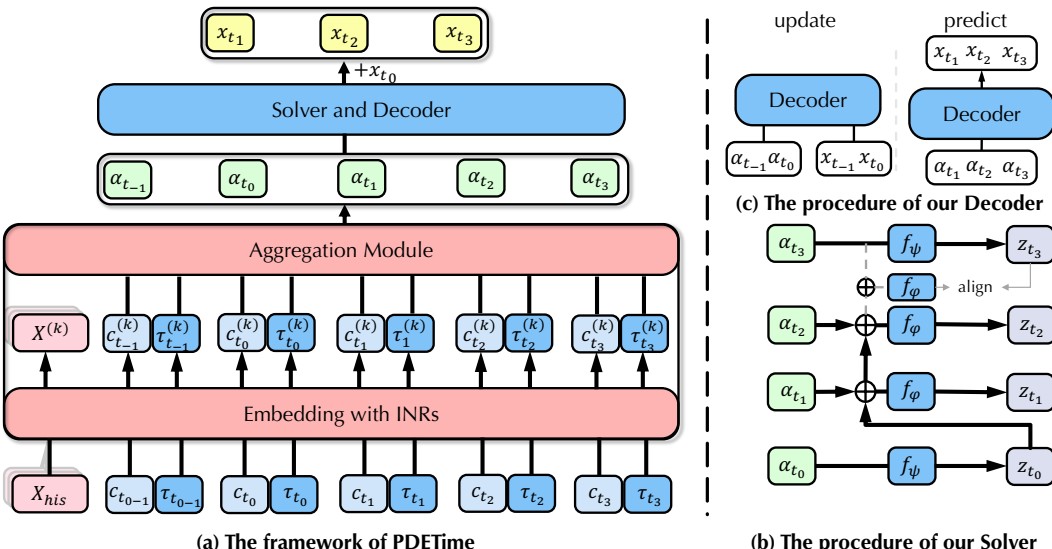

**(a) The framework of PDETime**

**(b) The procedure of our Solver**

**(c) The procedure of our Decoder**

Figure 2: The framework of proposed PDETime which consists of an Encoder $E_\theta$, a Solver, and a Decoder $D_\phi$. Given the initial condition $x_{t_0}$, PDETime first simulates $\frac{\partial \mathbf{u}(\mathbf{s},t)}{\partial t}$ at each time step $t$ using the Encoder $E_\theta(\mathbf{X}_{his}, \mathbf{c}_t, \tau_t)$; then uses the Solver to compute $\int_{t_0}^{t} \frac{\partial \mathbf{u}(\mathbf{s},\mu)}{\partial \mu} d\mu$, which is a numerical solver; finally, the Decoder maps integral term $\mathbf{z}_t$ from latent space to the value space and predict the final results $\hat{\mathbf{x}}_t = \mathbf{x}_{t_0} + D_\phi(\mathbf{z}_t)$.

$[\sin(2\pi\mathbf{b}_1\tau_t), \cos(2\pi\mathbf{b}_1\tau_t), ..., \sin(2\pi\mathbf{b}_v\tau_t), \cos(2\pi\mathbf{b}_v\tau_t)] \in \mathbb{R}^{vd}$, where $\mathbf{b}_v \in \mathbb{R}^{\frac{d}{2}}$ is sampled from $\mathcal{N}(0, 2^v)$.

After representing $\boldsymbol{\tau}_t^{(k)} \in \mathbb{R}^d$, $\mathbf{c}_t^{(k)} \in \mathbb{R}^o$, and $\mathbf{X}^{(k)} \in \mathbb{R}^{d \times C}$ with INRs, the Encoder aggregates $\mathbf{X}^{(k)}$ and $\mathbf{c}_t^{(k)}$ using $\boldsymbol{\tau}_t^{(k)}$ through the following equations:

$$\mathbf{s} = \text{LayerNorm}(\sum_{i=1}^{C} \frac{\boldsymbol{\tau}_t^{(k)} \cdot X^{(k)^i}}{\sum_{i=1}^{C} \boldsymbol{\tau}_t^{(k)} \cdot \mathbf{X}^{(K)^i}} X^{(k)^i} + \boldsymbol{\tau}_t^{(k)}),$$

$$\boldsymbol{\alpha}_t = \text{LayerNorm}(\mathbf{W}[\mathbf{s}; \mathbf{c}_t^{(k)}] + \mathbf{b} + \mathbf{s}), \tag{7}$$

where $[\cdot; \cdot]$ is the row-wise stacking operation. The aggregation process involves attention mechanisms (Vaswani et al., 2017) for spatial information and linear mapping for temporal information with $N$ layers. The complete pseudocode of the aggregation module is summarized in Appendix A.4.

Unlike previous works (Chen et al., 2018; Rubanova et al., 2019) which rely on the results of the previous steps, we directly compute $\boldsymbol{\alpha}_t$ at any time step, without the need for autoregressive calculation which can effectively alleviate the error accumulation problem and make the prediction results more stable (Brandstetter et al., 2021) and effectiveness.

### 3.2.2 SOLVER: $z_t = \int_{t_0}^{t} \alpha_\mu d\mu$

The Solver component introduces a numerical solver (Euler Solver) to compute the integral term $\mathbf{z}_t = \int_{t_0}^{t} \boldsymbol{\alpha}_\mu d\mu$, which can be approximated as:

$$\mathbf{z}_t = \int_{t_0}^{t} \frac{\partial \mathbf{u}(\mathbf{s}, \mu)}{\partial \mu} d\mu \approx \sum_{\mu=t_0}^{t} \frac{\partial u(x, \mu)}{\partial \mu} * \Delta\mu \approx \sum_{\mu=t_0}^{t} \boldsymbol{\alpha}_\mu * \Delta\mu, \tag{8}$$

where $t \in [0, H+L], t_0 = L$, and we set $\Delta\mu = 1$ for convenience. However, directly compute $\mathbf{z}_t = \sum_{\mu=t_0}^{t} \alpha_\mu * \Delta\mu$ through Eq 8 can easily lead to error accumulation and gradient problems (Rubanova

et al., 2019; Wu et al., 2022; Brandstetter et al., 2021) (also shown in our experimental results of Figure 3). To address these issues, we propose a modified solver that divides the time series sequence into non-overlapping patches of length $S$, where $\frac{H+L}{S}$ patches are obtained. For $t \bmod S = 0$, we directly estimate the integral term as $\mathbf{z}_t = f_\psi(\boldsymbol{\alpha}_t)$ using a neural network $f_\psi$. Otherwise, we use the numerical solver to estimate the integral term with the lower limit $\lfloor \frac{t}{S} \rfloor \cdot S$. This modification results in the following formula for the numerical solver:

$$\mathbf{z}_t = f_\psi(\boldsymbol{\alpha}_{t'}) + \int_{t'}^t f_\varphi(\boldsymbol{\alpha}_\mu)d\mu, \ \ t' = \lfloor \frac{t}{S} \rfloor * S, \tag{9}$$

where the neural networks $f_\psi$ and $f_\varphi$ are easily Linear layers. Furthermore, Eq 9 breaks the continuity and correlation between patches. To address this, we introduce an additional objective function $\mathcal{L}_c$ to ensure continuity and correlation as much as possible:

$$\mathcal{L}_c = \mathcal{L}(f_\psi(\boldsymbol{\alpha}_t), f_\psi(\boldsymbol{\alpha}_{t'}) + \int_{t'}^t f_\varphi(\boldsymbol{\alpha}_\mu)d\mu, \ \text{s.t.} \ t \bmod S = 0, \ t' = t - S. \tag{10}$$

We summarize the Solver as $z_t = \text{Solver}(\varphi, \psi, [\boldsymbol{\alpha}_{t_0}, ..., \boldsymbol{\alpha}_t], t_0, t)$ and the pseudocode of the Solver of PDETime is summarized in Appendix A.4.

### 3.2.3 DECODER: $\hat{x}_t = D_\phi(z_t) + x_{t_0}$

The Decoder component of our approach is responsible for decoding the estimated integral term $\mathbf{z}_t$ in the latent space back into the value space. As described in Eq 2, given the known initial condition $x_{t_0}$ (here we use the latest time step in the historical series as the initial condition), the Decoder predict the time step using the formula $\hat{x}_t = D_\phi(\mathbf{z}_t) + x_{t_0}$.

Following (Woo et al., 2023; Bertinetto et al., 2018), we also introduce meta-optimization to update the parameters in the Decoder to enhance the extrapolation capability of PDETime. Specifically, given the pair of look-back window $\mathbf{X}_{his} = [\mathbf{x}_{t_0-L+1}, ..., \mathbf{x}_{x_0}] \in \mathbb{R}^{L \times C}$ and horizon window $\mathbf{X}_{hor} = [\mathbf{x}_{t_0+1}, ..., \mathbf{x}_{t_0+H}] \in \mathbb{R}^{H \times C}$. We then use the parameters $\phi$ and $\theta, \varphi, \psi$ to adapt the look-back window and horizon window through a bi-level problem:

$$\phi^* = \arg\min_\phi \frac{1}{L} \sum_{t=t_0}^{t_0-L+1} \mathcal{L}_r(D_\phi(\text{Solver}(\varphi, \psi, [\boldsymbol{\alpha}_{t_0}, ..., \boldsymbol{\alpha}_t], t_0, t)), \mathbf{x}_t - \mathbf{x}_{t_0}), \tag{11}$$

$$\theta^*, \varphi^*, \psi^* = \arg\min_{\theta, \varphi, \psi} \frac{1}{H} \sum_{t=t_0+1}^{t_0+H} \mathcal{L}_p(D_\phi(\text{Solver}(\varphi, \psi, [\boldsymbol{\alpha}_{t_0}, ..., \boldsymbol{\alpha}_t], t_0, t)) + \mathbf{x}_{t_0}, \mathbf{x}_t), \tag{12}$$

where $\mathcal{L}_r$ and $\mathcal{L}_p$ denote the reconstruction and prediction loss, respectively (which will be described in detail in Section 3.3). During training, PDETime optimizes both $\theta$, $\psi$, $\varphi$, and $\phi$; while during inference, it only optimizes $\phi$ of Decoder to enhance the extrapolation. To ensure speed and efficiency, we employ the single ridge regression for $D_\phi$ (Bertinetto et al., 2018).

### 3.3 OPTIMIZATION

In Section 3.2.1, we discussed that it is challenging to ensure an exact match between $E_\theta(\mathbf{X}_{his}, \mathbf{c}_t, \tau_t)$ and $\frac{\partial \mathbf{u}(\mathbf{s},t)}{\partial t}$. To alleviate this problem, we introduce to achieve consistency between the first-order difference of the predicted sequence and target sequence with the additional optimization objective:

$$\mathcal{L}_f = \frac{1}{H} \sum_{t=t_0+1}^{t_0+H} \mathcal{L}(D_\phi(z_t) - D_\phi(z_{t-1}), x_t - x_{t-1}). \tag{13}$$

By minimizing $\mathcal{L}_f$, we encourage the first-order difference of the predicted sequence to match that of the target sequence. Additionally, when $\mathcal{L}_f \to 0$ and $\Delta t \to 0$, we observe that $\boldsymbol{\alpha}_t \propto \frac{\partial \mathbf{u}(\mathbf{s},t)}{\partial t}$. Furthermore, in Section 3.2.2, we set $\Delta t = 1$, leading to $\boldsymbol{\alpha}_t \propto \sum_{n=1}^\infty \frac{1}{n!} \frac{\partial^n \mathbf{u}(\mathbf{s},t)}{\partial t^n}$. In summary, $\boldsymbol{\alpha}_\mathbf{t}$ is related to the higher-order Taylor expansion of $\mathbf{u}(\mathbf{s},t)$ in the latent space (see more details in Appendix A.2), which ensures the stability of PDETime under discretization.

Combining $\mathcal{L}_p$, $\mathcal{L}_c$, and $\mathcal{L}_f$, the training objective becomes to:

$$\mathcal{L}_p = \mathcal{L}_p + \mathcal{L}_c + \mathcal{L}_f. \tag{14}$$

In the inference stage, we only need to minimize $\mathcal{L}_r$ which is the simple rescontruction loss.

## 4 EXPERIMENTS

### 4.1 EXPERIMENTAL SETTINGS

**Datasets.** We extensively include 7 real-world datasets in our experiments, including four ETT datasets (ETTh1, ETTh2, ETTm1, ETTm2) (Zhou et al., 2021). Electricity, Weather and Traffic (Wu et al., 2021), covering energy, transportation and weather domains (See Appendix A.1.1 for more details on the datasets). To ensure a fair evaluation, we follow the standard protocol of dividing each dataset into the training, validation and testing subsets according to the chronological order. The split ratio is 6:2:2 for the ETT dataset and 7:1:2 for the others (Zhou et al., 2021; Wu et al., 2021). We set the length of the lookback series as 512 for PatchTST, 336 for DLinear, and 96 for other historical-value-based models. The experimental settings of DeepTime remain consistent with the original settings (Woo et al., 2023). The prediction length varies in $\{96, 192, 336, 720\}$.

**Comparison methods.** We carefully choose 9 well-acknowledged historical-value-based models and 1 time-index-based model) as our benchmarks, including (1) Transformer-based models: FEDformer (Zhou et al., 2022), Stationary (Liu et al., 2022b), Crossformer (Zhang & Yan, 2023), PatchTST (Nie et al., 2023), and iTransformer (Liu et al., 2024); (2) Linear-based models: DLinear (Zeng et al., 2023) ; (3) CNN-based models: SCINet (Liu et al., 2022a), TimesNet (Wu et al., 2023); (4) Time-index-based model: DeepTime (Woo et al., 2023). (See Appendix A.1.2 for details of these baselines)

**Implementation Details.** Our method is trained with the Smooth L1 loss (Girshick, 2015) using the ADAM (Kingma & Ba, 2014) with the initial learning rate selected from $\{10^{-3}, 5 \times 10^{-4}, 10^{-4}\}$. Batch size is set to 32. All experiments are implemented in Pytorch (Paszke et al., 2019) and conducted on a single NVIDIA RTX 3090 GPUs with fixed feed 2024. Following DeepTime (Woo et al., 2023), we set the look-back length as $L = \mu * H$, where $\mu$ is a multiplier which decides the length of the look-back windows. We search through the values $\mu = [1, 3, 5, 7, 9]$, and select the best value based on the validation loss. We set layers of INRs $k = 5$ by default, and select the best results from $N = \{1, 2, 3, 5\}$. We summarize the temporal features used in this work in Appendix A.3.

### 4.2 MAIN RESULTS AND ABLATION STUDY

Comprehensive forecasting results are listed in Table 1 with the best in **Bold** and the second underlined. The lower MSE/MAE indicates the more accurate prediction result. Overall, PDETime achieves the best performance on most settings across seven real-world datasets compared with historical-value-based and time-index-based models. Additionally, experimental results also show that the performance of the proposed PDETime changes quite steadily as the prediction length $H$ increases. For instance, the MSE of PDETime increases from 0.330 to 0.365 on the Traffic dataset, while the MSE of PatchTST increases from 0.360 to 0.432, which is the SOTA historical-value-based model. This phenomenon was observed in other datasets and settings as well, indicating that PDETime retains better long-term robustness, which is meaningful for real-world practical applications.

We perform ablation studies on the Traffic and Weather datasets to validate the effect of **temporal feature** $c_t$, **spatial feature** $X_{his}$ and **initial condition** $x_{t_0}$. The results are presented in Table 2. **1)** The initial condition $x_{t_0}$ is useful on most settings. As mentioned in Section 3, we treat LMTF as Initial Value Problem, thus the effectiveness of $x_{t_0}$ validates the

Table 3: Analysis of the Solver and Initial value, w/o means discarding Solver and Initial value.

| Dataset | ETTh1 | | | | Weather | | | |
|---------|-------|------|------|------|---------|------|------|------|
| Model | PDETime | | w/o | | PDETime | | w/o | |
| Metric | MSE | MAE | MSE | MAE | MSE | MAE | MSE | MAE |
| 96 | 0.356 | 0.381 | 0.363 | 0.386 | 0.157 | 0.203 | 0.166 | 0.211 |
| 192 | 0.397 | 0.406 | 0.401 | 0.410 | 0.200 | 0.246 | 0.210 | 0.250 |
| 336 | 0.420 | 0.419 | 0.426 | 0.424 | 0.241 | 0.281 | 0.246 | 0.284 |
| 720 | 0.425 | 0.446 | 0.445 | 0.470 | 0.291 | 0.324 | 0.301 | 0.337 |

correctness of PDETime. **2)** The impact of spatial features $X_{his}$ on PDETime is limited. This may be due to the fact that the true spatial domains $s$ are unknown and complex, it is hard to utilize the historical observations $X_{his}$ to simulate $s$ with neural networks.

The spatial features $X_{his}$ are also beneficial in most cases, contributing to the stability of PDETime's performance. **3)** The influence of temporal feature $c_t$ on PDETime various significantly across different datasets. Experimental results have shown that $c_t$ is highly beneficial in the Traffic dataset, but its effect on Weather dataset is limited. For example, the period of Traffic dataset may be one day or one week, making it easier for PDETime to learn temporal features. On the other hand, the period

Table 1: Full results of the long-term forecasting task. We compare extensive competitive models under different prediction lengths following the setting of PatchTST (2023). The input sequence length is set to 336 and 512 for DLinear and PatchTST, and 96 for other historical-value-based baselines. Full results are listed in Table 7

| Models | PDETime (Ours) | | iTransformer (2024) | | PatchTST (2023) | | Crossformer (2023) | | DeepTime (2023) | | TimesNet (2023) | | DLinear (2023) | | SCINet (2022a) | | FEDformer (2022) | | Stationary (2022b) | |
|---|---|---|---|---|---|---|---|---|---|---|---|---|---|---|---|---|---|---|---|---|
| Metric | MSE | MAE | MSE | MAE | MSE | MAE | MSE | MAE | MSE | MAE | MSE | MAE | MSE | MAE | MSE | MAE | MSE | MAE | MSE | MAE |
| ETTm1 | **0.340** | **0.368** | 0.407 | 0.410 | 0.352 | 0.382 | 0.513 | 0.496 | 0.351 | 0.379 | 0.400 | 0.406 | 0.357 | 0.378 | 0.485 | 0.481 | 0.448 | 0.452 | 0.481 | 0.456 |
| ETTm2 | **0.241** | **0.295** | 0.288 | 0.332 | 0.256 | 0.316 | 0.757 | 0.610 | 0.262 | 0.326 | 0.291 | 0.333 | 0.267 | 0.331 | 0.571 | 0.537 | 0.305 | 0.349 | 0.306 | 0.347 |
| ETTh1 | **0.399** | **0.413** | 0.454 | 0.447 | 0.418 | 0.432 | 0.529 | 0.522 | 0.420 | 0.436 | 0.458 | 0.450 | 0.423 | 0.437 | 0.747 | 0.647 | 0.440 | 0.460 | 0.570 | 0.537 |
| ETTh2 | **0.334** | **0.379** | 0.383 | 0.407 | 0.343 | 0.387 | 0.942 | 0.684 | 0.489 | 0.472 | 0.414 | 0.427 | 0.431 | 0.446 | 0.954 | 0.723 | 0.437 | 0.449 | 0.526 | 0.516 |
| ECL | **0.150** | **0.244** | 0.178 | 0.270 | 0.159 | 0.252 | 0.244 | 0.334 | 0.164 | 0.265 | 0.192 | 0.295 | 0.166 | 0.263 | 0.268 | 0.365 | 0.214 | 0.327 | 0.193 | 0.296 |
| Traffic | **0.342** | **0.236** | 0.428 | 0.282 | 0.390 | 0.263 | 0.550 | 0.304 | 0.414 | 0.287 | 0.620 | 0.336 | 0.433 | 0.295 | 0.804 | 0.509 | 0.610 | 0.376 | 0.624 | 0.340 |
| Weather | **0.222** | **0.263** | 0.258 | 0.279 | 0.225 | 0.263 | 0.259 | 0.315 | 0.231 | 0.286 | 0.259 | 0.287 | 0.246 | 0.300 | 0.292 | 0.363 | 0.309 | 0.360 | 0.288 | 0.314 |
| 1st Count | 14 | 14 | 0 | 0 | 0 | 0 | 0 | 0 | 0 | 0 | 0 | 0 | 0 | 0 | 0 | 0 | 0 | 0 | 0 | 0 |

Table 2: Ablation study on variants of PDETime. -Temporal refers that removing the temporal domain feature $\mathbf{c}_t$; -Spatial refers that removing the historical observations $\mathbf{X}_{his}$; - Initial refers that removing the initial condition $\mathbf{x}_{t_0}$. The best results are highlighted in **bold**.

| Dataset | Models / Metric | PDETime MSE | PDETime MAE | -Temporal MSE | -Temporal MAE | -Spatial MSE | -Spatial MAE | -Initial MSE | -Initial MAE | -Temporal -Spatial MSE | -Temporal -Spatial MAE | - All MSE | - All MAE |
|---|---|---|---|---|---|---|---|---|---|---|---|---|---|
| Traffic | 96 | 0.330 | **0.232** | 0.336 | 0.236 | **0.329** | **0.232** | 0.334 | 0.235 | 0.394 | 0.268 | 0.401 | 0.269 |
| | 192 | **0.332** | **0.232** | 0.368 | 0.247 | 0.336 | 0.234 | 0.334 | **0.232** | 0.407 | 0.269 | 0.413 | 0.270 |
| | 336 | **0.342** | **0.236** | 0.378 | 0.251 | 0.344 | **0.236** | 0.343 | **0.236** | 0.419 | 0.273 | 0.426 | 0.272 |
| | 720 | **0.365** | **0.244** | 0.406 | 0.265 | 0.371 | 0.250 | 0.368 | 0.250 | 0.453 | 0.291 | 0.671 | 0.406 |
| Weather | 96 | **0.157** | **0.203** | 0.158 | 0.205 | 0.159 | 0.205 | 0.169 | 0.213 | 0.159 | 0.205 | 0.166 | 0.212 |
| | 192 | 0.200 | 0.246 | 0.206 | 0.253 | **0.198** | **0.243** | 0.208 | 0.248 | **0.198** | **0.243** | 0.208 | 0.250 |
| | 336 | 0.241 | 0.281 | **0.240** | 0.278 | 0.246 | 0.282 | 0.245 | 0.287 | **0.240** | **0.277** | 0.244 | 0.283 |
| | 720 | 0.291 | 0.324 | 0.292 | 0.323 | **0.290** | **0.322** | 0.300 | 0.337 | 0.294 | 0.327 | 0.299 | 0.337 |

of Weather dataset may be one year or longer, but the dataset only contains one year of data. As a result, PDETime cannot capture the complete temporal features in this case.

As mentioned in Sec 1, instead of directly utilizing neural networks, we aim to predict future series using Eq 2. To evaluate the effectiveness of this approach, we conduct experiments where PDETime can directly predict the target series by discarding $\mathbf{x}_{t_0}$ and the Solver. The experimental results, presented in Table 3, reveal that predicting future series with Eq 2 does indeed enhance the performance of PDETime. Additionally, we find that incorporating the Solver and $\mathbf{x}_t$ significantly improves the performance of time-index-based models, particularly when $\mathbf{X}_{his}$ and $\mathbf{c}_t$ are excluded (see details in Table 8). This further demonstrates the effectiveness of both the Solver and $\mathbf{x}_t$.

We conduct an additional ablation study on Traffic to evaluate the ability of different INRs to extract features of $\mathbf{X}_{his}$, $\mathbf{c}_t$, and $\tau_t$. In this study, we compared the performance of using the GELU or Tanh activation function instead of sine in SIREN and making $\tau_t^{(0)} = [\text{GELU}(2\pi\mathbf{b}_1\tau_t), \text{GELU}(2\pi\mathbf{b}_1\tau_t), ...]$ or $\tau^{(0)} = [\text{Tanh}(2\pi\mathbf{b}_1\tau_t), \text{Tanh}(2\pi\mathbf{b}_1\tau_t), ...]$. Table 5 presents the experimental results, we observe that the sine function (periodic functions) can extract features better than other non-decreasing activation functions. This is because the smooth, non-periodic activation functions fail to accurately model high-frequency information (Sitzmann et al., 2020). Time series data is often periodic, and the periodic nature of the sine function makes it more effective in extracting time series features.

### 4.3 EFFECTS OF HYPER-PARAMETERS

We evaluate the effect of four hyper-parameters: look-back window $L$, number of INRs layers $k$, number of aggregation layers $N$, and patch length $S$ on the ETTh1 and ETTh2 datasets. First, we perform a sensitivity on the look-back window $L = \mu * H$, where $H$ is based on the experimental setting. The results are presented in Table 4. We observe that the test error decreases

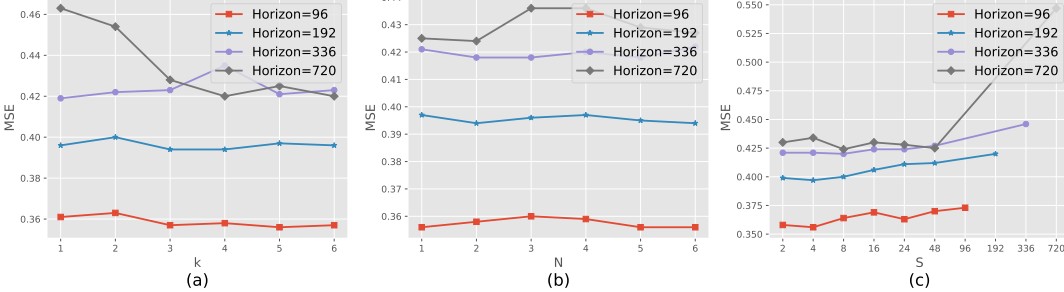

(a)                       (b)                       (c)

Figure 3: Evaluation on hyper-parameter impact. (a) MSE against hyper-parameter layers of INRs $k$ in Forecaster on ETTh1. (b) MSE against hyper-parameter layers of aggregation module $N$ in Forecaster on ETTh1. (c) MSE against hyper-parameter patch length $S$ in Estimator on ETTh1.

as $\mu$ increases, plateauing and even increasing slightly as $\mu$ grows extremely large when the horizon window is small. However, under a large horizon window, the test error increases as $\mu$ increases. Next, we evaluate the hyper-parameters $N$ and $k$ on PDETime, as shown in Figure 3 (a) and (b) respectively. We find that the performance of PDETime remains stable when $k \geq 3$. Additionally, the number of aggregation layers $N$ has a limited impact on PDETime.

Furthermore, we investigate the effect of patch length $S$ on PDETime, as illustrated in Figure 3 (c). We varied the patch length from 2 to 48 and evaluate MSE with different horizon windows. As the patch length $S$ increased, the prediction accuracy of PDETime initially improved, reached a peak, and then started to decline. However, the accuracy remains relatively stable throughout. We also extended the patch length to $S = H$. In this case, PDETime performed poorly, indicating that the accumulation of errors has a significant impact on the performance of PDETime. Overall, these

Table 4: Analysis on the look-back window length, based on the multiplier on horizon length, $L = \mu * H$. The best results are highlighted in **bold**.

| Dataset | Horizon | 96 | | 192 | | 336 | | 720 | |
|---|---|---|---|---|---|---|---|---|---|
| | $\mu$ | MSE | MAE | MSE | MAE | MSE | MAE | MSE | MAE |
| ETTh1 | 1 | 0.378 | 0.386 | 0.415 | 0.411 | **0.421** | **0.420** | **0.425** | **0.446** |
| | 3 | 0.359 | 0.382 | **0.394** | **0.404** | 0.427 | 0.421 | 0.443 | 0.460 |
| | 5 | 0.360 | 0.385 | 0.396 | 0.405 | **0.421** | **0.420** | 0.495 | 0.501 |
| | 7 | **0.354** | **0.381** | 0.398 | 0.405 | 0.427 | 0.429 | 0.545 | 0.532 |
| | 9 | 0.356 | **0.381** | 0.397 | 0.406 | 0.446 | 0.440 | 1.220 | 0.882 |
| ETTh2 | 1 | 0.288 | 0.335 | 0.357 | 0.381 | 0.380 | 0.404 | **0.380** | **0.421** |
| | 3 | 0.276 | 0.331 | 0.339 | 0.374 | **0.358** | **0.395** | 0.422 | 0.456 |
| | 5 | 0.275 | 0.333 | **0.331** | **0.370** | 0.360 | 0.408 | 0.622 | 0.576 |
| | 7 | **0.268** | **0.330** | **0.331** | 0.378 | 0.384 | 0.427 | 0.624 | 0.595 |
| | 9 | 0.272 | 0.331 | **0.331** | 0.378 | 0.412 | 0.451 | 0.797 | 0.689 |

analyses provide insights into the effects of different hyper-parameters on the performance of PDE-Time and can guide the selection of appropriate settings for achieving optimal results.

To address potential concern regarding the inclusion of additional temporal information in our method, we conducted comprehensive experiments comparing PDETime with TiDE which also utilizes dynamic covariates and PatchTST. In order to ensure a fair comparison, we also augmented PatchTST with temporal information. The results in Table 10 reveal that even with the inclusion of temporal information, TiDE and PatchTST still exhibit weaker performance compared to

Table 5: Analysis on INRs. PDETime refers to our proposed approach. GELU and Tanh refer to replacing SIREN and CFF with GELU or Tanh activation, respectively. The best results are highlighted in **bold**.

| Dataset | Method | PDETime | | GELU | | Tanh | |
|---|---|---|---|---|---|---|---|
| | Metric | MSE | MAE | MSE | MAE | MSE | MAE |
| Traffic | 96 | **0.330** | **0.232** | 0.332 | 0.237 | 0.338 | 0.233 |
| | 192 | **0.332** | **0.232** | 0.338 | 0.241 | 0.339 | 0.235 |
| | 336 | **0.342** | **0.236** | 0.348 | 0.244 | 0.348 | 0.238 |
| | 720 | **0.365** | **0.244** | 0.376 | 0.252 | 0.366 | **0.244** |

PDETime. We also conducted ablation studies to validate the effectiveness of Solver, initial conditions, as well as loss functions $l_r$ and $l_c$. The results of these experiments can be found in Appendix A.5. Additionally, due to space constraints, we provide visualizations and convergence experiments in Appendix A.6 and Appendix A.7, respectively.

## 5 CONCLUSION AND FUTURE WORK

In this paper, we propose a novel LMTS framework PDETime, based on neural Solvers, which consists of Encoder, Solver, and Decoder. Specifically, the Encoder simulates the temporal derivative in latent space in parallel. The solver is responsible for computing the integral term with improved stability. Finally, the Decoder maps the integral term from latent space into the value space and predicts the target series under the initial condition. Additionally, we incorporate meta-optimization techniques to enhance the ability of PDETime to extrapolate future series. Extensive experimental results show that PDETime achieves state-of-the-art performance across forecasting benchmarks on various real-world datasets. We also perform ablation studies to identify the key components contributing to the success of PDETime.

**Future Work.** Firstly, while our proposed neural solver, PDETime, has shown promising results for long-term multivariate time series forecasting, there are other types of neural solvers that could potentially be applied to this task. Exploring these alternative neural solvers and comparing their performance on LMTF could be an interesting future direction. Additionally, our PDETime have demonstrated strong capabilities in handling regular time series data. Therefore, another potential future direction is to apply PDETime to irregular time series tasks, such as missing value imputation. Finally, our approach of rethinking long-term multivariate time series forecasting from the perspective of partial differential equations has led to state-of-the-art performance, exploring other perspectives and frameworks to tackle this task could be a promising direction for future research.

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

# A   APPENDIX

In this section, we present the experimental details of PDETime. The organization of this section is as follows:

- Appendix A.1 provides details on the datasets and baselines.
- Appendix A.3 provides details of time features used in this work.
- Appendix A.4 provides pseudocode of Encoder, Solver, and Training procedure of PDETime.
- Appendix A.5 presents the results of the robustness experiments and full results of Table 1.
- Appendix A.6 visualizes the prediction results of PDETime on seven real-world datasets.
- Appendix A.7 visualizes the Training, validation, and test losses of seven real-world datasets.

## A.1   EXPERIMENTAL DETAILS

### A.1.1   DATASETS

we use the most popular multivariate datasets in LMTF, including ETT, Electricity, Traffic and Weather:

- The ETT (Zhou et al., 2021) (Electricity Transformer Temperature) dataset contains two years of data from two separate countries in China with intervals of 1-hour level (ETTh) and 15-minute level (ETTm) collected from electricity transformers. Each time step contains six power load features and oil temperature.
- The Electricity [1] dataset describes 321 clients' hourly electricity consumption from 2012 to 2014.
- The Traffic [2] dataset contains the road occupancy rates from various sensors on San Francisco Bay area freeways, which is provided by California Department of Transportation.
- the Weather [3] dataset contains 21 meteorological indicators collected at around 1,600 landmarks in the United States.

Table 6 presents key characteristics of the seven datasets. The dimensions of each dataset range from 7 to 862, with frequencies ranging from 10 minutes to 7 days. The length of the datasets varies from 966 to 69,680 data points. We split all datasets into training, validation, and test sets in chronological order, using a ratio of 6:2:2 for the ETT dataset and 7:1:2 for the remaining datasets.

Table 6: Statics of Dataset Characteristics.

| Datasets | ETTh1 | ETTh2 | ETTm1 | ETTm2 | Electricity | Traffic | Weather |
|---|---|---|---|---|---|---|---|
| Dimension | 7 | 7 | 7 | 7 | 321 | 862 | 21 |
| Frequency | 1 hour | 1 hour | 15 min | 15 min | 1 hour | 1 hour | 10 min |
| Length | 17420 | 17420 | 69680 | 69680 | 26304 | 52696 | 17544 |

[1] https://archive.ics.uci.edu/ml/datasets/ElectricityLoadDiagrams20112014.
[2] http://pems.dot.ca.gov.
[3] https://www.bgc-jena.mpg.de/wetter/.

### A.1.2 BASELINES

We choose SOTA and the most representative LMTF models as our baselines, including historical-value-based and time-index-based models, as follows:

- PatchTST Nie et al. (2023): the current historical-value-based SOTA models. It utilizes channel-independent and patch techniques and achieves the highest performance by utilizing the native Transformer.

- DLinear Zeng et al. (2023): a highly insightful work that employs simple linear models and trend decomposition techniques, outperforming all Transformer-based models at the time.

- Crossformer Zhang & Yan (2023): similar to PatchTST, it utilizes the patch technique commonly used in the CV domain. However, unlike PatchTST's independent channel design, it leverages cross-dimension dependency to enhance LMTF performance.

- FEDformer Zhou et al. (2022): it employs trend decomposition and Fourier transformation techniques to improve the performance of Transformer-based models in LMTF. It was the best-performing Transformer-based model before Dlinear.

- Stationary Liu et al. (2022b): it proposes a De-stationary Attention to alleviate the over-stationarization problem.

- iTransformer Liu et al. (2024): it different from pervious works that embed multivariate points of each time step as a (temporal) token, it embeds the whole time series of each variate independently into a (variate) token, which is the extreme case of Patching.

- TimesNet Wu et al. (2023): it transforms the 1D time series into a set of 2D tensors based on multiple periods and uses a parameter-efficient inception block to analyze time series.

- SCINet Liu et al. (2022a): it proposes a recursive downsample-convolve-interact architecture to aggregate multiple resolution features with complex temporal dynamics.

- DeepTime Woo et al. (2023): it is the first time-index-based model in long-term multivariate time-series forecasting.

### A.2 $\alpha_t \propto \frac{\partial u(t)}{\partial t}$ WITH $\mathcal{L}_f \to 0$ AND $\Delta t \to 0$

We first assume that $\mathcal{L}_f \to 0$, then we have:

$$
\begin{aligned}
\lim_{\mathcal{L}_f \to 0} \mathbf{u}(\mathbf{s}, t_1) - \mathbf{u}(\mathbf{s}, t_0) &= D_\phi(\mathbf{z}_{t_1}) - D_\phi(\mathbf{z}_{t_0}) \\
&= D_\phi(\mathbf{z}_{t_1} - \mathbf{z}_{t_0}) \\
&= D_\phi(\boldsymbol{\alpha}_{t_0} * \Delta t)
\end{aligned}
\tag{15}
$$

With Taylor expansion, we have

$$
\begin{aligned}
\lim_{\mathcal{L}_f \to 0} D_\phi(\boldsymbol{\alpha}_{t_0} * \Delta t) &= \mathbf{u}(\mathbf{s}, t_1) - \mathbf{u}(\mathbf{s}, t_0) \\
&= \frac{\partial \mathbf{u}(\mathbf{s}, t_0)}{\partial t_0} dt + \frac{\partial \mathbf{u}(\mathbf{s}, t_0)}{\partial s} ds + ... + \sum_{i=1}^{n} \frac{1}{n!} \frac{\partial^n \mathbf{u}(\mathbf{s}, t_0)}{\partial \mathbf{s}^i \partial t_0^{n-i}} d\mathbf{x}^i dt^{n-i} \\
&= \frac{\partial \mathbf{u}(\mathbf{s}, t_0)}{\partial t_0} dt + \frac{1}{2} \frac{\partial^2 \mathbf{u}(\mathbf{s}, t_0)}{\partial t_0^2} dt^2 + ... + \frac{1}{n!} \frac{\partial^n \mathbf{u}(\mathbf{s}, t_0)}{\partial t_0^n} dt^n \\
&= \sum_{n=1}^{\infty} \frac{1}{n!} \frac{\partial^n \mathbf{u}(\mathbf{s}, t_0)}{\partial t_0^n} dt^n
\end{aligned}
\tag{16}
$$

We assume that $W_\phi \geq 0$ in $D_\phi$, and $\Delta t \to 0$ then

$$
\lim_{\mathcal{L}_f \to 0} \lim_{\Delta t \to 0} D_\phi(\boldsymbol{\alpha}_{t_0} * \Delta t) = \frac{\partial \mathbf{u}(\mathbf{s}, t_0)}{\partial t_0} * \Delta t + \frac{\partial^2 \mathbf{u}(\mathbf{s}, t_0)}{\partial t_0^2} * (\Delta t)^2 + \mathcal{O}((\Delta t)^3)
\tag{17}
$$

$$
\approx \frac{\partial \mathbf{u}(\mathbf{s}, t)}{\partial t} * \Delta t
\tag{18}
$$

$$
\lim_{\mathcal{L}_f \to 0} \lim_{\Delta t \to 0} \alpha_{t_0} \propto \frac{\partial u(x, t_0)}{\partial t_0}
\tag{19}
$$

In our paper, we set $\Delta t = 1$, thus we have

$$\lim_{\mathcal{L}_f} \boldsymbol{\alpha}_t \propto \sum_{n=1}^{\infty} \frac{1}{n!} \frac{\partial^n \mathbf{u}(\mathbf{s}, t)}{\partial t^n} \tag{20}$$

In this case, $\boldsymbol{\alpha}_t$ is related to the higher-order Taylor expansion of $\mathbf{u}(\mathbf{s}, t)$ in the latent space, thus we can predict $x_{t_1} = \mathbf{u}(\mathbf{s}, t_1) = \mathbf{u}(\mathbf{s}, t_0) + D_\phi(\boldsymbol{\alpha}_{t_0})$.

### A.3 TEMPORAL FEATURES

Depending on the sampling frequency, the temporal feature $t_\tau$ of each dataset is also different. We will introduce the temporal feature of each data set in detail:

- ETTm and Weather: day-of-year, month-of-year, day-of-week, hour-of-day, minute-of-hour.
- ETTh, Traffic, and Electricity: day-of-year, month-of-year, day-of-week, hour-of-day.

we also normalize these features into [0,1] range.

### A.4 PSEUDOCODE

We provide the pseudo-code of Encoder and Solver in Algorithms 1 and Algorithms 2. We also provide the training procedure of PDETime in Algorithm 3

---

**Algorithm 1** Pseudocode of the aggregation module of Encoder

---

**Input:** Time-index feature $\boldsymbol{\tau}_t^{(k)}$, temporal feature $\mathbf{c}_t^{(k)}$ and historical feature $\mathbf{X}^{(k)}$.

1: $\boldsymbol{\tau}_t^{(k)}, \mathbf{c}_t^{(k)}, \mathbf{X}^{(k)} = \mathbf{W}_\tau^1 \boldsymbol{\tau}_t^{(k)} + \mathbf{b}_\tau^1, \mathbf{W}_c^1 \mathbf{c}_t^{(k)} + \mathbf{b}_c^1, \mathbf{W}_x^1 \mathbf{X}^{(k)} + \mathbf{b}_x^1 \qquad \triangleright \boldsymbol{\tau}_t \in \mathbb{R}^d, \mathbf{X} \in \mathbb{R}^{d \times C},$
   $\mathbf{c}_t \in \mathbb{R}^o$

2: $\boldsymbol{\tau}_t^{(k)}, \mathbf{c}_t^{(k)}, \mathbf{X}^{(k)} = \text{LayerNorm}(\text{GeLU}(\boldsymbol{\tau}_t^{(k)})), \text{LayerNorm}(\sin(\mathbf{c}_t^{(k)})), \text{LayerNorm}(\text{GeLU}(\mathbf{X}^{(k)}))$

3: $\mathbf{s} = \text{LayerNorm}(\sum_{i=1}^{C} \frac{\boldsymbol{\tau}_t^{(k)} \cdot \mathbf{X}^{(k)i}}{\sum_{i=1}^{C} \boldsymbol{\tau}_t^{(k)} \cdot \mathbf{X}^{(k)i}} + \boldsymbol{\tau}_t^{(k)})$

4: $\mathbf{s} = \mathbf{W}^1[\mathbf{s}; \mathbf{c}_t^{(k)}] + \mathbf{b}^1 + \mathbf{s}$

5: $\mathbf{s} = \text{LayerNorm}(\mathbf{s})$

6: **for** $n = 2, ..., N$ **do**

7: $\quad \mathbf{s}, \mathbf{c}_t^{(k)}, \mathbf{X}^{(k)} = \mathbf{W}_s^n \mathbf{s} + \mathbf{b}_s^n, \mathbf{W}_c^n \mathbf{c}_t^{(k)} + \mathbf{b}_c^n, \mathbf{W}_x^n \mathbf{X}^{(k)} + \mathbf{b}_x^n$

8: $\quad \mathbf{s}, \mathbf{c}_t^{(k)}, \mathbf{X}^{(k)} = \text{LayerNorm}(\text{GeLU}(\tau_t^{(k)})), \text{LayerNorm}(\sin(\mathbf{c}_t^{(k)})), \text{LayerNorm}(\text{GeLU}(\mathbf{X}^{(k)}))$

9: $\quad \mathbf{s} = \text{LayerNorm}(\sum_{i=1}^{C} \frac{\mathbf{s} \cdot \mathbf{X}^{(k)i}}{\sum_{i=1}^{C} \mathbf{s} \cdot \mathbf{X}^{(k)i}} + \mathbf{s})$

10: $\quad \mathbf{s} = \mathbf{W}^n[\mathbf{s}; \mathbf{c}_t^{(k)}] + \mathbf{b}^n + \mathbf{s}$

11: $\quad \mathbf{s} = \text{LayerNorm}(\mathbf{s})$

12: **end for**

13: $\boldsymbol{\alpha}_t \leftarrow \mathbf{s}$

14: **return** $\boldsymbol{\alpha}_t$ $\hfill \triangleright \boldsymbol{\alpha}_t \in \mathbb{R}^d$

---

**Algorithm 2** Solver of PDETime

---

**Input:** latent partial derivative $[\boldsymbol{\alpha}_{t_0}, ..., \boldsymbol{\alpha}_t]$ lower limit $t_0$, upper limit $t$, and patch length $S$.

1: **if** $t \mod S = 0$ **then**

2: $\quad \mathbf{z}_t \leftarrow f_\psi(\boldsymbol{\alpha}_t)$

3: **else**

4: $\quad t' \leftarrow t' \leftarrow \lfloor \frac{t}{S} \rfloor$

5: $\quad \mathbf{z}_t \leftarrow f_\psi + \sum_{\mu=t'}^{t} f_\varphi(\boldsymbol{\alpha}_\mu) * \Delta\mu$

6: **end if**

7: **return** $\mathbf{z}_t$ $\hfill \triangleright \mathbf{z}_t \in \mathbb{R}^d$

---

---

**Algorithm 3** Training procedure of PDETime

---

**Input:** Model $E_\theta$, $f_\psi$, $f_\varphi$ and $D_\phi$ with parameters $\theta$, $\psi$, $\varphi$, and $\phi$
**Input:** Learning rates $\eta$
1: **for** $e$ in epochs **do**
2:     **for** $s$ in samples **do**
3:         **for** $t = t_{0-L+1}, ..., t_0, ..., t_{0+H}$ **do**
4:             $\alpha_t \leftarrow E_\theta(\mathbf{X}_{his}, \mathbf{c}_t, \tau_t)$
5:         **end for**
6:         **for** $t = t_{0-L+1}, ..., t_0, ..., t_{0+H}$ **do**
7:             $\mathbf{z}_i \leftarrow \text{Solver}(\varphi, \psi, [\boldsymbol{\alpha}_{t_0}, ..., \boldsymbol{\alpha}_t], t_0, t)$
8:         **end for**
9:         $\mathbf{Z}_{his}, \mathbf{Z}_{hor} \leftarrow [\mathbf{z}_{t_{0-L+1}}, ..., \mathbf{z}_{t_0}], [\mathbf{z}_{t_{0+1}}, ..., \mathbf{z}_{t+H}]$
10:        $\phi \leftarrow (\mathbf{Z}_{his}^T \mathbf{Z}_{his} + \lambda I)^{-1} \mathbf{Z}_{his}^T (\mathbf{X}_{his} - \mathbf{x}_{t_0})$
11:        $\hat{\mathbf{X}}_{hor} \leftarrow D_\phi(\mathbf{Z}_{hor}) + \mathbf{x}_{t_0}$
12:        compute training loss $\mathcal{L}_p$ with Eq. 14
13:        $\theta \leftarrow \theta - \eta \nabla_\theta \mathcal{L}_p$
14:        $\psi \leftarrow \psi - \eta \nabla_\psi \mathcal{L}_p$
15:        $\varphi \leftarrow \varphi - \eta \nabla_\varphi \mathcal{L}_p$
16:     **end for**
17: **end for**

---

## A.5 EXPERIMENTAL RESULTS OF ROBUSTNESS

The experimental results of the robustness of our algorithm based on Solver and Initial condition are summarized in Table 8. We also test the effectiveness of continuity loss $\mathcal{L}_c$ and $\mathcal{L}_r$ in Table 9. The experimental results in Table 8 demonstrate that PDETime can achieve strong performance on the ETT dataset even when using only the Solver or initial value conditions, without explicitly incorporating spatial and temporal information. Moreover, combining the initial value conditions with the Solver further enhances the performance of PDETime. These findings suggest that PDETime exhibits promising capabilities and can perform well even in scenarios with limited data availability. Additionally, we conducted an analysis on the ETTh1 and ETTh2 datasets to investigate the impact of the loss term $\mathcal{L}_c$ and $\mathcal{L}_r$. Our findings demonstrate that incorporating $\mathcal{L}_c$ into PDETime can enhance its robustness. In addition, we also find that loss $\mathcal{L}_r$ has a large impact on the effectiveness of our model, which demonstrates the importance of extrapolation capability to PDETime.

To address potential concerns regarding the inclusion of additional temporal information in our method, we conducted comprehensive experiments comparing PDETime with existing approaches, including TiDE (also utilizes dynamic covariates) and PatchTST. In order to ensure a fair comparison, we also augmented PatchTST with temporal information. The experimental results, presented in Table 10, reveal that even with the inclusion of temporal information, TiDE and PatchTST still exhibit weaker performance compared to PDETime. Notably, directly incorporating temporal information into PatchTST led to a significant performance degration. These findings highlight the importance of a well-designed and purposeful integration of temporal features.

## A.6 VISUALIZATION

We visualize the prediction results of PDETime on seven real-world datasets. As illustrated in Figure 4, for prediction lengths $H = 96, 192, 336, 720$, the prediction curve closely aligns with the ground-truth curves in most cases (except for the weather dataset, which we suspect that weather forecasting is more difficult than other domains) , indicating the outstanding predictive performance of PDETime. Meanwhile, PDETime demonstrates effectiveness in capturing periods of time features.

## A.7 CONVERGENCE

We conducted additional experiments to validate the convergence property of PDETime. Figure 5 illustrates the training, validation, and test loss of our model as the number of epochs increases. It is evident that all losses initially decrease and then plateau. Notably, the training losses of ETTh2 and ETTm2 exhibit significant fluctuations, while the validation and test losses remain consistently stable. We speculate that this behavior may be attributed to the relatively small scale of the ETTh2 and ETTm2 datasets. Conversely, for large-scale datasets such as Traffic and Electricity, all losses, including training, validation, and test, demonstrate remarkable stability.

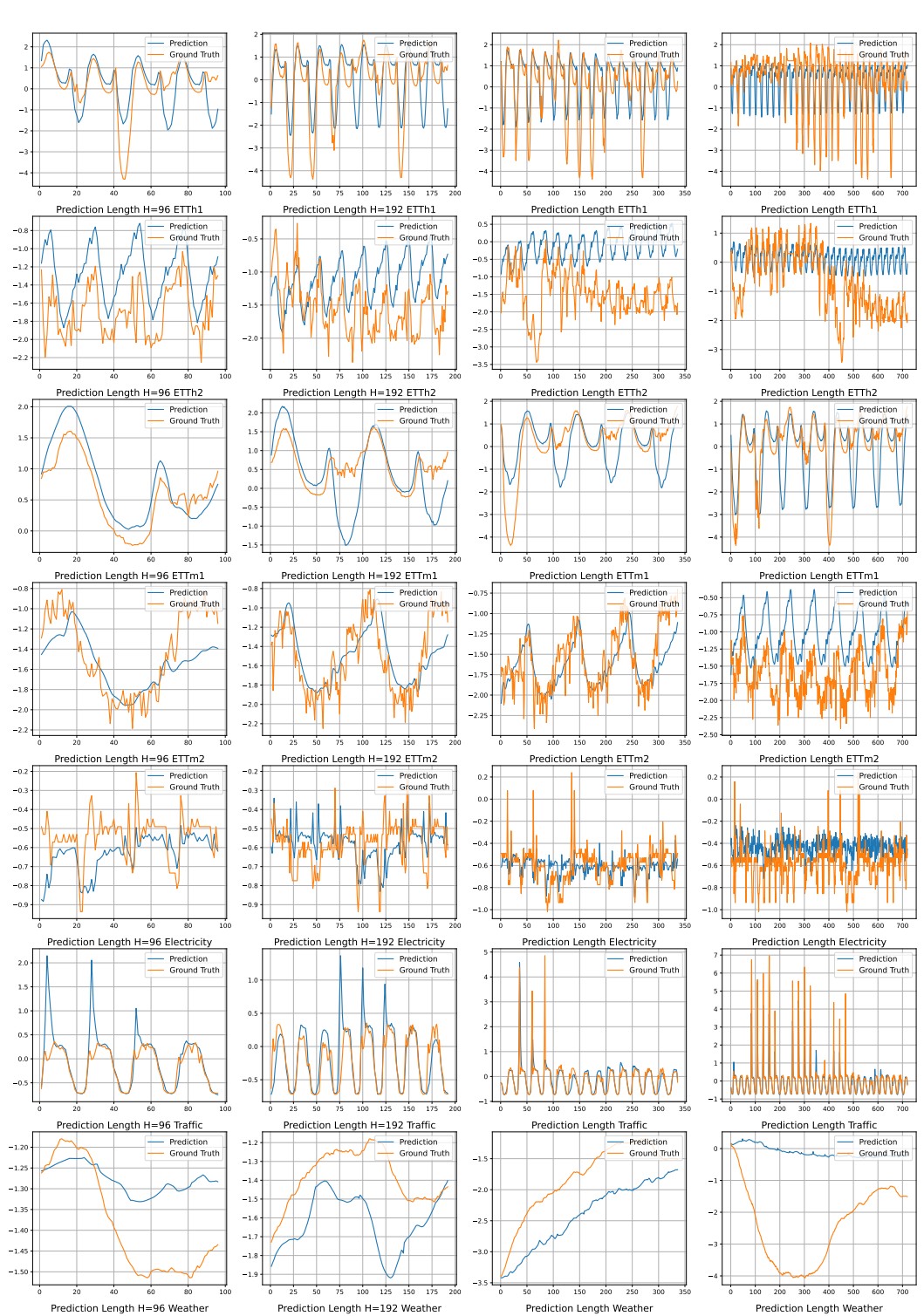

Figure 4: Visualization of long-term forecasting results on seven datasets.

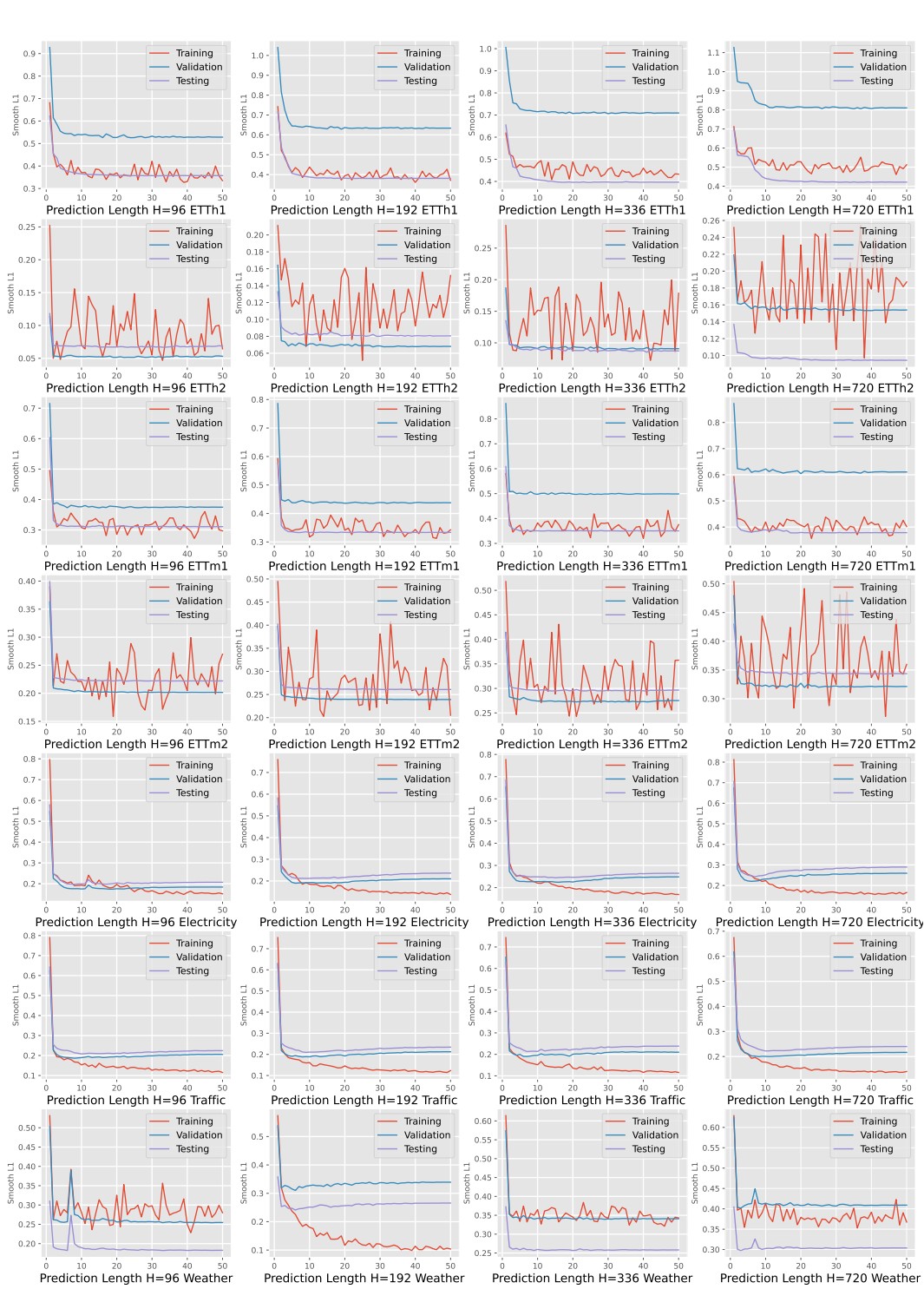

Figure 5: Training, validation, and test losses changes for 50 epochs.

Table 7: Full results of the long-term forecasting task. We compare extensive competitive models under different prediction lengths following the setting of PatchTST (2023). The input sequence length is set to 336 and 512 for DLinear and PatchTST, and 96 for other historical-value-based baselines. *Avg* means the average results from all four prediction lengths.

| Models | PDETime (Ours) | | iTransformer (2024) | | PatchTST (2023) | | Crossformer (2023) | | DeepTime (2023) | | TimesNet (2023) | | DLinear (2023) | | SCINet (2022a) | | FEDformer (2022) | | Stationary (2022b) | |
|---|---|---|---|---|---|---|---|---|---|---|---|---|---|---|---|---|---|---|---|---|
| Metric | MSE | MAE | MSE | MAE | MSE | MAE | MSE | MAE | MSE | MAE | MSE | MAE | MSE | MAE | MSE | MAE | MSE | MAE | MSE | MAE |
| ETTm1 96 | **0.292** | **0.335** | 0.334 | 0.368 | 0.293 | 0.346 | 0.404 | 0.426 | 0.305 | 0.347 | 0.338 | 0.375 | 0.299 | 0.343 | 0.418 | 0.438 | 0.379 | 0.419 | 0.386 | 0.398 |
| ETTm1 192 | **0.329** | **0.359** | 0.377 | 0.391 | 0.333 | 0.370 | 0.450 | 0.451 | 0.340 | 0.371 | 0.374 | 0.387 | 0.335 | 0.365 | 0.439 | 0.450 | 0.426 | 0.441 | 0.459 | 0.444 |
| ETTm1 336 | **0.346** | **0.374** | 0.426 | 0.420 | 0.369 | 0.392 | 0.532 | 0.515 | 0.362 | 0.387 | 0.410 | 0.411 | 0.369 | 0.386 | 0.490 | 0.485 | 0.445 | 0.459 | 0.495 | 0.464 |
| ETTm1 720 | **0.395** | **0.404** | 0.491 | 0.459 | 0.416 | 0.420 | 0.666 | 0.589 | 0.399 | 0.414 | 0.478 | 0.450 | 0.425 | 0.421 | 0.595 | 0.550 | 0.543 | 0.490 | 0.585 | 0.516 |
| ETTm1 Avg | **0.340** | **0.368** | 0.407 | 0.410 | 0.352 | 0.382 | 0.513 | 0.496 | 0.351 | 0.379 | 0.400 | 0.406 | 0.357 | 0.378 | 0.485 | 0.481 | 0.448 | 0.452 | 0.481 | 0.456 |
| ETTm2 96 | **0.158** | **0.244** | 0.180 | 0.264 | 0.166 | 0.256 | 0.287 | 0.366 | 0.166 | 0.257 | 0.187 | 0.267 | 0.167 | 0.260 | 0.286 | 0.377 | 0.203 | 0.287 | 0.192 | 0.274 |
| ETTm2 192 | **0.213** | **0.283** | 0.250 | 0.309 | 0.223 | 0.296 | 0.414 | 0.492 | 0.225 | 0.302 | 0.249 | 0.309 | 0.224 | 0.303 | 0.399 | 0.445 | 0.269 | 0.328 | 0.280 | 0.339 |
| ETTm2 336 | **0.262** | **0.318** | 0.311 | 0.348 | 0.274 | 0.329 | 0.597 | 0.542 | 0.277 | 0.336 | 0.321 | 0.351 | 0.281 | 0.342 | 0.637 | 0.591 | 0.325 | 0.366 | 0.334 | 0.361 |
| ETTm2 720 | **0.334** | **0.336** | 0.412 | 0.407 | 0.362 | 0.385 | 1.730 | 1.042 | 0.383 | 0.409 | 0.408 | 0.403 | 0.397 | 0.421 | 0.960 | 0.735 | 0.421 | 0.415 | 0.417 | 0.413 |
| ETTm2 Avg | **0.241** | **0.295** | 0.288 | 0.332 | 0.256 | 0.316 | 0.757 | 0.610 | 0.262 | 0.326 | 0.291 | 0.333 | 0.267 | 0.331 | 0.571 | 0.537 | 0.305 | 0.349 | 0.306 | 0.347 |
| ETTh1 96 | **0.356** | **0.381** | 0.386 | 0.405 | 0.379 | 0.401 | 0.423 | 0.448 | 0.371 | 0.396 | 0.384 | 0.402 | 0.375 | 0.399 | 0.654 | 0.599 | 0.376 | 0.419 | 0.513 | 0.491 |
| ETTh1 192 | **0.397** | **0.406** | 0.441 | 0.436 | 0.413 | 0.429 | 0.471 | 0.474 | 0.403 | 0.420 | 0.436 | 0.429 | 0.405 | 0.416 | 0.719 | 0.631 | 0.420 | 0.448 | 0.534 | 0.504 |
| ETTh1 336 | **0.420** | **0.419** | 0.487 | 0.458 | 0.435 | 0.436 | 0.570 | 0.546 | 0.433 | 0.436 | 0.491 | 0.469 | 0.439 | 0.443 | 0.778 | 0.659 | 0.459 | 0.465 | 0.588 | 0.535 |
| ETTh1 720 | **0.425** | **0.446** | 0.503 | 0.491 | 0.446 | 0.464 | 0.653 | 0.621 | 0.474 | 0.492 | 0.521 | 0.500 | 0.472 | 0.490 | 0.836 | 0.699 | 0.506 | 0.507 | 0.643 | 0.616 |
| ETTh1 Avg | **0.399** | **0.413** | 0.454 | 0.447 | 0.418 | 0.432 | 0.529 | 0.522 | 0.420 | 0.436 | 0.458 | 0.450 | 0.423 | 0.437 | 0.747 | 0.647 | 0.440 | 0.460 | 0.570 | 0.537 |
| ETTh2 96 | **0.268** | **0.330** | 0.297 | 0.349 | 0.274 | 0.335 | 0.745 | 0.584 | 0.287 | 0.352 | 0.340 | 0.374 | 0.289 | 0.353 | 0.707 | 0.621 | 0.358 | 0.397 | 0.476 | 0.458 |
| ETTh2 192 | **0.331** | **0.370** | 0.380 | 0.400 | 0.342 | 0.382 | 0.877 | 0.656 | 0.383 | 0.412 | 0.402 | 0.414 | 0.383 | 0.418 | 0.860 | 0.689 | 0.429 | 0.439 | 0.512 | 0.493 |
| ETTh2 336 | **0.358** | **0.395** | 0.428 | 0.432 | 0.365 | 0.404 | 1.043 | 0.731 | 0.523 | 0.501 | 0.452 | 0.452 | 0.448 | 0.465 | 1.000 | 0.744 | 0.496 | 0.487 | 0.552 | 0.551 |
| ETTh2 720 | **0.380** | **0.421** | 0.427 | 0.445 | 0.393 | 0.430 | 1.104 | 0.763 | 0.765 | 0.624 | 0.462 | 0.468 | 0.605 | 0.551 | 1.249 | 0.838 | 0.463 | 0.474 | 0.562 | 0.560 |
| ETTh2 Avg | **0.334** | **0.379** | 0.383 | 0.407 | 0.343 | 0.387 | 0.942 | 0.684 | 0.489 | 0.472 | 0.414 | 0.427 | 0.431 | 0.446 | 0.954 | 0.723 | 0.437 | 0.449 | 0.526 | 0.516 |
| ECL 96 | **0.129** | **0.222** | 0.148 | 0.240 | 0.129 | 0.222 | 0.219 | 0.314 | 0.137 | 0.238 | 0.168 | 0.272 | 0.140 | 0.237 | 0.247 | 0.345 | 0.193 | 0.308 | 0.169 | 0.273 |
| ECL 192 | **0.143** | **0.235** | 0.162 | 0.253 | 0.147 | 0.240 | 0.231 | 0.322 | 0.152 | 0.252 | 0.184 | 0.289 | 0.153 | 0.249 | 0.257 | 0.355 | 0.201 | 0.315 | 0.182 | 0.286 |
| ECL 336 | **0.152** | **0.248** | 0.178 | 0.269 | 0.163 | 0.259 | 0.246 | 0.337 | 0.166 | 0.268 | 0.198 | 0.300 | 0.169 | 0.267 | 0.269 | 0.369 | 0.214 | 0.329 | 0.200 | 0.304 |
| ECL 720 | **0.176** | **0.272** | 0.225 | 0.317 | 0.197 | 0.290 | 0.280 | 0.363 | 0.201 | 0.302 | 0.220 | 0.320 | 0.203 | 0.301 | 0.299 | 0.390 | 0.246 | 0.355 | 0.222 | 0.321 |
| ECL Avg | **0.150** | **0.244** | 0.178 | 0.270 | 0.159 | 0.252 | 0.244 | 0.334 | 0.164 | 0.265 | 0.192 | 0.295 | 0.166 | 0.263 | 0.268 | 0.365 | 0.214 | 0.327 | 0.193 | 0.296 |
| Traffic 96 | **0.330** | **0.232** | 0.395 | 0.268 | 0.360 | 0.249 | 0.522 | 0.290 | 0.390 | 0.275 | 0.593 | 0.321 | 0.410 | 0.282 | 0.788 | 0.499 | 0.587 | 0.366 | 0.612 | 0.338 |
| Traffic 192 | **0.332** | **0.232** | 0.417 | 0.276 | 0.379 | 0.256 | 0.530 | 0.293 | 0.402 | 0.278 | 0.617 | 0.336 | 0.423 | 0.287 | 0.789 | 0.505 | 0.604 | 0.373 | 0.613 | 0.340 |
| Traffic 336 | **0.342** | **0.236** | 0.433 | 0.283 | 0.392 | 0.264 | 0.558 | 0.305 | 0.415 | 0.288 | 0.629 | 0.336 | 0.436 | 0.296 | 0.797 | 0.508 | 0.621 | 0.383 | 0.618 | 0.328 |
| Traffic 720 | **0.365** | **0.244** | 0.467 | 0.302 | 0.432 | 0.286 | 0.589 | 0.328 | 0.449 | 0.307 | 0.640 | 0.350 | 0.466 | 0.315 | 0.841 | 0.523 | 0.626 | 0.382 | 0.653 | 0.355 |
| Traffic Avg | **0.342** | **0.236** | 0.428 | 0.282 | 0.390 | 0.263 | 0.550 | 0.304 | 0.414 | 0.287 | 0.620 | 0.336 | 0.433 | 0.295 | 0.804 | 0.509 | 0.610 | 0.376 | 0.624 | 0.340 |
| Weather 96 | 0.157 | 0.203 | 0.174 | 0.214 | **0.149** | **0.198** | 0.158 | 0.230 | 0.166 | 0.221 | 0.172 | 0.220 | 0.176 | 0.237 | 0.221 | 0.306 | 0.217 | 0.296 | 0.173 | 0.223 |
| Weather 192 | 0.200 | 0.246 | 0.221 | 0.254 | **0.194** | **0.241** | 0.206 | 0.277 | 0.207 | 0.261 | 0.219 | 0.261 | 0.220 | 0.282 | 0.261 | 0.340 | 0.276 | 0.336 | 0.245 | 0.285 |
| Weather 336 | **0.241** | **0.281** | 0.278 | 0.296 | 0.245 | 0.282 | 0.272 | 0.335 | 0.251 | 0.298 | 0.280 | 0.306 | 0.265 | 0.319 | 0.309 | 0.378 | 0.339 | 0.380 | 0.321 | 0.338 |
| Weather 720 | **0.291** | **0.324** | 0.358 | 0.349 | 0.314 | 0.334 | 0.398 | 0.418 | 0.301 | 0.338 | 0.365 | 0.359 | 0.323 | 0.362 | 0.377 | 0.427 | 0.403 | 0.428 | 0.414 | 0.410 |
| Weather Avg | **0.222** | **0.263** | 0.258 | 0.279 | 0.225 | 0.263 | 0.259 | 0.315 | 0.231 | 0.286 | 0.259 | 0.287 | 0.246 | 0.300 | 0.292 | 0.363 | 0.309 | 0.360 | 0.288 | 0.314 |
| 1st Count | 33 | 33 | 0 | 0 | 2 | 2 | 0 | 0 | 0 | 0 | 0 | 0 | 0 | 0 | 0 | 0 | 0 | 0 | 0 | 0 |

## A.8 LIMITATIONS

While PDETime represents a significant advancement in long-term multivariate time series forecasting with PDE solvers, it currently has limitations that should be addressed in future research. Firstly, PDETime is not well-suited for modeling irregular time series as it operates under the assumption that historical observations $\mathbf{X}_{his}$ are regular. However, PDETime can still predict irregular future data by modifying $\Delta t$. Secondly, PDETime considers spatial information $\mathbf{s}$ to be unknown and requires estimation through various well-designed neural networks. It is important to note that spatial information may be highly complex and challenging to predict directly using neural networks.

## A.9 BROADER IMPACTS

This paper presents PDETime, a new PDE-based method in Long-term multivariate time series forecasting. This paper only focuses on the algorithm design. Using all the codes and datasets strictly follows the corresponding licenses. There is no potential ethical risk or negative social impact.

Table 8: Analysis on Solver and Initial condition. INRs refers to only using INRs to represent $\tau_t$; + Initial refers to aggregating initial condition $\mathbf{x}_{t_0}$; +Solver refers to using numerical solvers to compute integral terms in latent space. The best results are highlighted in **bold**.

| Dataset | Method Metric | INRs MSE | MAE | INRs+Initial MSE | MAE | INRs+Solver MSE | MAE | INRs+Initial+Solvers MSE | MAE |
|---------|---------------|----------|-----|------------------|-----|-----------------|-----|--------------------------|-----|
| ETTh1 | 96 | 0.371 | 0.396 | 0.364 | 0.384 | 0.358 | 0.381 | 0.358 | 0.381 |
| | 192 | 0.403 | 0.420 | 0.402 | 0.409 | 0.398 | 0.407 | 0.397 | 0.406 |
| | 336 | 0.433 | 0.436 | 0.428 | 0.420 | 0.348 | 0.238 | 0.422 | 0.420 |
| | 720 | 0.474 | 0.492 | 0.439 | 0.452 | 0.455 | 0.476 | 0.437 | 0.450 |
| ETTh2 | 96 | 0.287 | 0.352 | 0.270 | 0.330 | 0.285 | 0.342 | 0.270 | 0.331 |
| | 192 | 0.383 | 0.412 | 0.331 | 0.372 | 0.345 | 0.379 | 0.329 | 0.369 |
| | 336 | 0.523 | 0.501 | 0.373 | 0.405 | 0.357 | 0.399 | 0.354 | 0.399 |
| | 720 | 0.765 | 0.624 | 0.392 | 0.429 | 0.412 | 0.444 | 0.395 | 0.428 |

Table 9: Analysis on the effectiveness of loss term $\mathcal{L}_c$ and $\mathcal{L}_r$.

| Dataset | Method Metric | PDETime MSE | MAE | PDETime-$\mathcal{L}_c$ MSE | MAE | PDETime-$\mathcal{L}_r$ MSE | MAE |
|---------|---------------|-------------|-----|-----------------------------|-----|-----------------------------|-----|
| ETTh1 | 96 | 0.356 | 0.381 | 0.357 | 0.381 | 0.740 | 0.598 |
| | 192 | 0.397 | 0.406 | 0.393 | 0.405 | 0.870 | 0.694 |
| | 336 | 0.420 | 0.419 | 0.422 | 0.420 | 0.688 | 0.557 |
| | 720 | 0.425 | 0.419 | 0.446 | 0.458 | 0.799 | 0.653 |
| ETTh2 | 96 | 0.268 | 0.330 | 0.271 | 0.330 | 0.431 | 0.423 |
| | 192 | 0.331 | 0.370 | 0.341 | 0.373 | 0.435 | 0.467 |
| | 336 | 0.358 | 0.395 | 0.363 | 0.397 | 0.426 | 0.460 |
| | 720 | 0.380 | 0.421 | 0.396 | 0.434 | 0.468 | 0.489 |

Table 10: Analysis on the effectiveness of Temporal Feature.

| Dataset | Method Metric | PDETime MSE | MAE | TiDE MSE | MAE | PatchTST MSE | MAE | PatchTST+ Temporal MSE | MAE |
|---------|---------------|-------------|-----|----------|-----|--------------|-----|------------------------|-----|
| ETTh1 | 96 | 0.356 | 0.335 | 0.375 | 0.398 | 0.379 | 0.401 | 0.378 | 0.403 |
| | 192 | 0.397 | 0.406 | 0.412 | 0.422 | 0.413 | 0.429 | 0.414 | 0.425 |
| | 336 | 0.420 | 0.419 | 0.435 | 0.433 | 0.435 | 0.436 | 0.449 | 0.449 |
| | 720 | 0.425 | 0.446 | 0.454 | 0.465 | 0.446 | 0.464 | 0.507 | 0.499 |
| ETTh2 | 96 | 0.268 | 0.330 | 0.270 | 0.336 | 0.274 | 0.335 | 0.323 | 0.376 |
| | 192 | 0.331 | 0.370 | 0.332 | 0.380 | 0.342 | 0.382 | 0.375 | 0.416 |
| | 336 | 0.358 | 0.395 | 0.360 | 0.407 | 0.365 | 0.404 | 0.400 | 0.430 |
| | 720 | 0.380 | 0.421 | 0.419 | 0.451 | 0.393 | 0.430 | 0.428 | 0.454 |

