# OpenReview forum: "PDETime: Rethinking Long-term Multivariate Time Series Forecasting from the Perspective of Partial Differential Equations"
_ICLR.cc/2025/Conference — Submitted to ICLR 2025_

### Official Review · Reviewer_ySQw · 2024-10-16

**Soundness:** 3
**Presentation:** 3
**Contribution:** 3
**Rating:** 6
**Confidence:** 5

**Summary:**

This paper proposes a novel method for multivariate time series forecast by treating the input continuous dynamical system governed by a latent PDE equation. Instead of predicting future values directly, PDETime predicts the derivative and uses Euler method to integrate the derivative, and applies encoder and decoder for input embedding and yielding the final output. This approach enhances both performance
and stability, especially in scenarios with long-horizon forecast. Extensive experiments on seven diverse real-world LMTF datasets demonstrate that PDETime not only adapts effectively to the intrinsic spatiotemporal characteristics.

**Strengths:**

The key idea of this paper is novel: the author introduces a PDE solver to solve time series instead of using historical data as input, which may be able to capture the inter-series relationship for enhanced accuracy. The paper is also well written with solid experiments.

**Weaknesses:**

The literature review setion seems a little over-compressed, especially the multivariate time series literature. In fact, there is much more literature in the past 2 years that seeks to provide forecast from multiple perspectives. For example, Transformer-based methods, linear interpolation methods (D-linear) and auxiliary series construction (CATS: Enhancing Multivariate Time Series Forecasting by Constructing Auxiliary Time Series as Exogenous Variables, ICML 2024) and so on. I would suggest that the author should enrich their literature on multivariate time series, and compress the neural PDE literature because it is only utilized as a component of the model.

Also the benchmark methods are not sufficient as there are many papers in 2024 that address LMTF from LLM perspectiuve, including LLM4TS, GPT4TS, CATS and so on. I strongly encourage the author to add these benchmarks into their comparison.

I would gladly increase my rating if my concerns are addressed.

**Questions:**

I have several questions about this paper:
1. Why does the author think multivariate time series must have a latent PDE structure? For example, ETTH data does not seem to be governed by any PDE dynamics.
2. The PDETime seems more like ODETime to me, as the author only calculates time derivative instead of spatial derivative.
3. It is well known that the numerical integration step can be time consuming. I wonder how the model performs with smaller Euler steps in both forecast accuracy and time consumption.
4. I wonder what is the purpose of utilizing encoder and decoder in this problem along with PDE solver. Does it change the input dimention to facilitate numerical integration?
5. Why does the author utilizes both sin() and GeLU() function? Is their any mathematical intuitions behind this?

---

### Official Review · Reviewer_B2Vh · 2024-10-31

**Soundness:** 2
**Presentation:** 3
**Contribution:** 2
**Rating:** 6
**Confidence:** 3

**Summary:**

The paper provides a novel approach, PDETime, for long-term multivariate time-series forecasting (LMTF) by modeling time series data as samples from a continuous dynamical system governed by partial differential equations. PDETime introduces an encoding-integration-decoding architecture to predict future values in the latent space by estimating the partial derivative of the system with respect to time. The method employs a neural solver to address the nonstationarity and sampling limitations common in traditional historical-value- and time-index-based models. Finally, the author provides experimental results demonstrating that PDETime achieves state-of-the-art performance across seven benchmark datasets. In my opinion, the paper is organized and easy to follow, but I have some questions.

**Strengths:**

1. The paper presents a unique approach for time-series forecasting by framing the task as an initial value problem governed by PDEs, which provides a theoretically grounded approach for capturing spatiotemporal dependencies.
2. The architecture effectively leverages the latent space to model complex temporal dependencies, mitigating issues of error accumulation that are common in autoregressive methods.
3. Extensive evaluations across seven real-world datasets reveal that PDETime outperforms leading historical-value-based and time-index-based models, including Transformer and CNN models, demonstrating robust, stable performance over varying forecasting lengths.
4. The study includes comprehensive ablation experiments on key components (e.g., Solver, temporal and spatial features) and evaluates the impact of hyper-parameters, enhancing the reproducibility and scientific rigor of the results.

**Weaknesses:**

1. The use of numerical solvers (e.g., Euler Solver) could introduce sensitivity to hyper-parameter tuning and potential limitations in modeling high-frequency or abrupt temporal changes.
2. The encoding-integration-decoding process, along with meta-optimization, may add computational overhead, especially for large datasets or models with high-dimensional data, limiting scalability.

**Questions:**

1. In Section 3.2.2, is the solution generated by the (modified) Euler solver guaranteed to be bounded? Furthermore, as the Euler method is known to be potentially unstable and sensitive to step size, could the authors clarify how they determined the appropriate step size for this implementation? In other words, does it raise any issue by setting $\Delta \mu=1$?

2. Following previous question 1, could authors provide some theoretical or empirical evidence of the stability and boundedness of their modified Euler solver? Additionally, could the author discuss any potential limitations or tradeoffs of using $\Delta \mu=1$ compared to other step sizes.

3. In the Conclusions and Future Works section, the authors note that future developments may enhance PDETime’s ability to handle irregular time series or data with missing values. Could the authors further elaborate on the current limitations of PDETime in this regard?

4. Could the authors provide a detailed analysis of the time and space complexity of PDETime compared to the baseline methods? Given that the method combines aspects of both historical-value- and time-index-based models, does this lead to an increase in complexity compared to other models?

---

### Official Review · Reviewer_3cLh · 2024-11-01

**Soundness:** 2
**Presentation:** 1
**Contribution:** 2
**Rating:** 3
**Confidence:** 3

**Summary:**

The authors propose a novel model, PDETime, for long-term multivariate time series forecasting. PDETime assumes that the multivariate time series is sampled from a continuous dynamical system governed by a partial differential equation (PDE). Unlike existing approaches that rely solely on historical observations or discrete time points, PDETime leverages both observations and time for forecasting.

PDETime encodes the partial derivatives of the system, solves them for future time points in a latent space using the proposed PDE solver, and then decodes the results back to the data space with a decoder. Experiments conducted on seven multivariate datasets with varying forecast horizons show significant improvement in forecasting accuracy compared to baseline models.

**Strengths:**

**S1** Viewing multivariate time series as instances of partial differential equations (PDEs) is an intriguing approach. This perspective is particularly relevant for multivariate time series in which sensors at different locations measure the same thing (e.g., electricity data).

**S2** Authors proposed a new PDE solver which can potentially avoid the problems with Neural-ODE solvers

**S2** The experimental results on various datasets indicate that the approach outperforms a range of baselines in multivariate longterm forecasting. Various ablation studies have been shown in the experiments

**Weaknesses:**

**W1. Writing Quality:**

1. The paper contains undefined variables when first introduced. E.g. $\mathbf{X}{his}$, $\mathbf{c}{t}$, and $\tau_t$ in line 8, as well as $\mathbf{W}{\tau}$, $\mathbf{W}c$, $\mathbf{W}x$, $\mathbf{W}$, $\mathbf{b}\tau$, $\mathbf{b}c$, $\mathbf{b}x$, and $\mathbf{b}$. Please define any variable when it is first introduced.
2. Section 3.1 has an incorrect sub-section title. The section is titled "Problem Formulation" but describes the model's overview. Problem formulation typically refers to the specific problem the paper addresses.
3. Some statements are overly vague. For instance, in lines 502-505, "Finally, our approach of rethinking long-term ... promising direction for future research" lacks clarity. Please mention what new perspectives do the authors wish to explore?
4. Please review the order of all tables in the paper; Table 1 should appear before Table 3.
5. Typos:

5.1 line 52: hiders $\rightarrow$ hinders

5.2 line 294: $\textbf{x}{x_0} \rightarrow \textbf{x}{t_0}$

5.3 line 151: $\textbf{u}(t + \Delta) \rightarrow \textbf{u}(t + \Delta t)$

5.4 lines 77-80: "Furthermore, as shown in .... capture temporal dependencies" first sentence doesn't complete the thought

5.5 Eq. 10, $\mathcal{L}($: "(" did not close

5.6 Alg 2, line 4: $t'$ appeared twice

6. There are many other notational issues which require careful attention from authors.
ex. time index have subscript at some places like eq. 2 and superscript in eq. 1

**W2.** Motivation for considering multivariate time series as an instance of PDE is missing. Can authors provide clear motivation behind this approach? Would this assumption hold for multivariate time series data with variables measuring different aspects, such as a physiological dataset where one variable tracks pulse and another tracks blood pressure?

**W3. Model Scope:**

1. Why is the model limited to long-horizon predictions? How does it perform on short horizons?
2. The authors state that existing models rely on either historical observations or time points alone. However, the Informer model uses both local and global time information as embeddings. Can authors clarify this?

**W3. Encoder Architecture:**

1. What is the motivation behind using this particular encoder architecture?
2. The encoder’s design is challenging to understand due to insufficient explanations:
- Clearly define and distinguish between $X^{(k)^i}$ and $\mathbf{X}^{(K)^i}$
- Provide a detailed explanation of the operations in Equation 7
- Explain the rationale behind adding $\mathbf{c}_t^{(k)}$ again after concatenation
- Explain the discrepancy between Alg 1 and Eq 7 (see line 3, Alg 1)

3. Eq. 6; why (6a) and (6b) LHS have subscript $t$ and no information of it on RHS?
4. What is For loop in Alg 1? Is it clearly mentioned in main text?

**W4. Solver:**

1. The paper does not clearly explain how the proposed solver circumvents the issues associated with the Euler solver. Please provide a comparison between their proposed solver and the Euler/Neural-ODE solver
2. Figure 2b does not effectively illustrate the solver. A clearer figure would help.
3. The PDE solver is claimed to be a novel and key contribution. To substantiate this, could the authors demonstrate the advantages of this solver over existing Neural ODE solvers (e.g., Chen et al., 2018) through a toy example?
4. In line 270, the authors state that Figure 3 illustrates the solver’s advantages over the Euler solver. However, it is difficult to see this comparison, as Figure 3 primarily shows results for hyperparameters $k$, $N$, and $S$.

**W5. Experiments:**

1. Please clarify the input and prediction sequence lengths for the results in Table 1. Comparing Tables 1 and 7, it appears the forecasting horizon is 336 (please correct me if this is incorrect)
2. There is a discrepancy in the third decimal place of Patch TST results between these tables if the forecasting horizon is 336
3. Please inform about Table 7 in paragraph after Section 4.2, otherwise the text is confusing

**Questions:**

See weaknesses

---

### Official Review · Reviewer_fPTa · 2024-11-01

**Soundness:** 3
**Presentation:** 3
**Contribution:** 4
**Rating:** 6
**Confidence:** 4

**Summary:**

In this paper, the authors propose PDETime, a novel model for time-series
forecasting. The methods novelty lies in the fact, that it does not simply
predict values for a specific time point, but that it generates an encoding
$\alpha_t$ which can be interpreted as a first deviation. Then, the dynamics
of the hidden representation $z_t$ of a time in the interval $[t_0,t] $ are computed
via an integral solver over $\int_{t_0,t} \alpha_t $ and this is then used to decode via $\text{Decoder}(z_t) + x_{t_0}$.

**Strengths:**

+ The idea of capturing the time-dynamics and treating time-series forecasting as an initial-condition problem is a cool idea.
+ The results are promising, PDETime always outperforms all competitors
+ The approach could be potential starting point for a new direction in time-series forecasting

**Weaknesses:**

- Please write a clear problem formulation: In Time-Series forecasting, what do
  you have given, what do you want to predict/which objective do you want to
  optimize? What are the domains your inputs and outputs live in. I had to
  somehow guess here sometimes at the beginning, especially it was not clear to
  me at the beginning what $s$ is, and whether it is given or not.
- I really like the PDE motivation, but what you at the end do is simply to
  learn first derivations at time points instead of learning the values
  them self. As your spatial input $s$ is computed in the encoder and you are
  only considering the derivation with respect to time for fixed s, one could
  argue that at the end you are only considering ODEs with respect to time.
  However, then what you do is already established in the literature for
  time-series forecasting with irregular time-points, see [1], [2], [3].

[1] De Brouwer, E., Simm, J., Arany, A., & Moreau, Y. (2019). GRU-ODE-Bayes: Continuous modeling of sporadically-observed time series. Advances in neural information processing systems, 32.

[2]Scholz, Randolf, et al. "Latent Linear ODEs with Neural Kalman Filtering for Irregular Time Series Forecasting." (2023).

[3] Schirmer, Mona, et al. "Modeling irregular time series with continuous recurrent units." International conference on machine learning. PMLR, 2022.
- As your approach is fundamentally different to established transformer models,
  it is important to integrate a runtime-study. What does the performance gain
  over PatchTST cost with respect to efficiency?

**Questions:**

## Main Questions Connected to Mentioned Weaknesses

- What is the key difference in your approach compared to the NeuralODE methods mentioned under weaknesses? Please do give an answer which goes beyond "We are coming from PDEs not ODEs", because at the end you only consider deviations with respect to time.
- What is runtime-behavior with respect to the considered baselines?

## Further Questions/Comments
- I was wondering what the forecasting horizon is in Table 1. I had to look at Table 7 in the appendix to see that is is an average over different horizons. I would suggest to put that information into the caption of Table 1.
- What are the results if you replace (9) with always using$ f_{\psi}(z_t)?$
- Please explicitly write down the reconstruction loss L_p
- What are the input time features c^(0) in general/For the particular datasets used?

## Some Small Typos/Comments
- Page 6, line 294: X_{x_0} -> X_{t_0}
- (14): I guess, here somethings is wrong, as L_p occurs on both side of the equation

---

### Official Review · Reviewer_tFF4 · 2024-11-04

**Soundness:** 3
**Presentation:** 2
**Contribution:** 2
**Rating:** 3
**Confidence:** 3

**Summary:**

The paper presents PDETime, a novel approach for long-term multivariate time-series forecasting that models the series as a continuous dynamical system governed by partial differential equations (PDEs). Instead of directly forecasting future values, PDETime predicts the partial derivatives in the latent space, integrating this information over time to generate forecasts.

**Strengths:**

1. The application of PDEs in time-series forecasting introduces a unique perspective for capturing continuous dynamical patterns.
2. The authors test PDETime on a wide range of datasets, demonstrating the model’s adaptability.

**Weaknesses:**

1. **Fairness in Comparison**: My most concern is that the experiments may lack fairness due to differences in historical input length (\(H\)) between PDETime and baseline models. While PDETime’s \(H\) is optimized, baseline models use a fixed input length, which can skew results since (1) different input lengths impact the number of samples in the test set for each model, potentially affecting comparability, and (2) input length significantly influences forecasting performance. It’s recommended to either standardize \(H\) across models or optimize it for all baselines.
2. **Code Availability**: The absence of released code reduces the credibility and reproducibility of the results, as reviewers and readers cannot verify the findings independently.
3. **Unclear Loss Weighting**: The weight description for the loss function in Equation 14 is unclear, making it challenging to understand the balance between different loss components.
4. **Lack of Efficiency Analysis**: There is no discussion of PDETime’s computational efficiency, such as runtime or memory consumption, which is essential for understanding its scalability and practicality for long-term forecasting tasks.

**Questions:**

1. Can the authors clarify the weighting mechanism in Equation 14 for a more transparent understanding of how different losses are balanced?
2. Would it be possible to release the code or provide more detailed implementation details to support reproducibility?
3. How does PDETime perform in terms of runtime and memory usage compared to other baseline models, especially for very long forecasting windows?

---

### Meta-Review · Area_Chair_ckcW · 2024-12-11

**Metareview:**

This paper introduces PDETime, a novel approach to long-term multivariate time-series forecasting using partial differential equations (PDEs) to model the temporal dynamics of latent representations. While the perspective of treating time series as samples from a PDE is intriguing, the paper has critical issues that limit its impact and clarity. The work lacks sufficient empirical rigor, with comparisons to established baselines potentially biased due to differences in input length and insufficient benchmarking against recent advanced methods. While the PDE-based modeling is positioned as novel, the approach seems more akin to ordinary differential equations (ODEs), with limited exploration of spatial derivatives, reducing the distinctiveness of the method. Additionally, concerns about the computational efficiency and scalability of the proposed solver are unaddressed, raising questions about its practicality for real-world use. The presentation has several weaknesses, including unclear problem formulation, inconsistent notation, and an over-reliance on vague claims rather than explicit justifications. While the experimental results appear promising, the lack of code availability and detailed runtime analyses further hinders reproducibility and assessment. These shortcomings, combined with the overextension of the claimed contributions, justify a decision of Reject.

**Additional Comments On Reviewer Discussion:**

N/A

---

### Decision · Program_Chairs · 2025-01-22

Reject